# Dengue genetic divergence generates within-serotype antigenic variation, but serotypes dominate evolutionary dynamics

Sidney M Bell[1,2], Leah Katzelnick[3,4], Trevor Bedford[1]*

[1]Vaccine and Infectious Disease Division, Fred Hutchinson Cancer Research Center, Seattle, United States; [2]Molecular and Cell Biology Program, University of Washington, Seattle, United States; [3]Division of Infectious Diseases and Vaccinology, University of California, Berkeley, Berkeley, United States; [4]Department of Biology, University of Florida, Gainesville, United States

**Abstract** Dengue virus (DENV) exists as four genetically distinct serotypes, each of which is historically assumed to be antigenically uniform. Recent analyses suggest that antigenic heterogeneity may exist within each serotype, but its source, extent and impact remain unclear. Here, we construct a sequence-based model to directly map antigenic change to underlying genetic divergence. We identify 49 specific substitutions and four colinear substitution clusters that robustly predict dengue antigenic relationships. We report moderate antigenic diversity within each serotype, resulting in genotype-specific patterns of heterotypic cross-neutralization. We also quantify the impact of antigenic variation on real-world DENV population dynamics, and find that serotype-level antigenic fitness is a dominant driver of dengue clade turnover. These results provide a more nuanced understanding of the relationship between dengue genetic and antigenic evolution, and quantify the effect of antigenic fitness on dengue evolutionary dynamics.

*For correspondence:
tbedford@fredhutch.org

Competing interests: The authors declare that no competing interests exist.

## Introduction

Dengue virus (DENV) is a mosquito-borne flavivirus which consists of four genetically distinct clades, canonically thought of as serotypes (DENV1 – DENV4) (*Lanciotti et al., 1997*). DENV circulates primarily in South America and Southeast Asia, infecting 400 million people annually. Primary DENV infection is more often mild and is thought to generate lifelong homotypic immunity and temporary heterotypic immunity, which typically wanes over 6 months to 2 years (*Sabin, 1952*; *Reich et al., 2013*; *Katzelnick et al., 2016*). Subsequent heterotypic secondary infection induces broad cross-protection, and symptomatic tertiary and quaternary cases are rare (*Gibbons et al., 2007*; *Olkowski et al., 2013*). However, a small subset of secondary infections are enhanced by non-neutralizing, cross-reactive antibodies, resulting in severe disease via antibody-dependent enhancement (ADE) (*Halstead, 1979*; *Katzelnick et al., 2017*; *Sangkawibha et al., 1984*; *Salje et al., 2018*). Approximately 1–3% of cases progress to severe dengue hemorrhagic fever, causing ∼ 9000 deaths each year (*Bhatt et al., 2013*; *Stanaway et al., 2016*) and relative risk of severe dengue from secondary heterotypic infection relative to primary infection is estimated to be ∼ 24 (*Mizumoto et al., 2014*). Thus, the antigenic relationships between dengue viruses — describing whether the immune response generated after primary infection results in protection or enhancement of secondary infection — are key drivers of DENV case outcomes and epidemic patterns.

While each serotype is clearly genetically and antigenically distinct, it is not clear how subserotype clades of DENV interact antigenically. Each DENV serotype consists of broad genetic diversity

(*Figure 1A*), including canonical clades termed 'genotypes' (*Rico-Hesse, 1990*; *Twiddy et al., 2003*). Specific genotypes have been associated with characteristically mild or severe disease, and heterogeneous neutralization titers suggest that the immune response to some genotypes is more cross-protective than others (*Gentry et al., 1982*; *Russell and Nisalak, 1967*). Until recently, it has been assumed that these intraserotype differences are minimally important compared to interserotype differences. However, empirical evidence has demonstrated that these genotype-specific differences can drive case outcomes and epidemic severity (reviewed in *Holmes and Twiddy, 2003*). For example, analysis of a longitudinal cohort study demonstrated that specific combinations of primary infection serotype and secondary infection genotype can mediate individual case outcomes (*OhAinle et al., 2011*). On a population scale, the DENV1-immune population of Iquitos, Peru, experienced either entirely asymptomatic or very severe epidemic seasons in response to two different genotypes of DENV2 (*Kochel et al., 2002*).

One explanation for these and similar observations is that overlooked intraserotype antigenic variation contributes to these genotype-specific case outcomes and epidemic patterns. Recent efforts to antigenically characterize diverse DENV viruses suggests that each serotype may contain antigenic heterogeneity, but the source and impact of this heterogeneity is not clear (*Katzelnick et al., 2015*). Here, we characterize the evolutionary basis for observed antigenic heterogeneity among DENV clades. We also quantify the impact of within- and between-serotype antigenic variation on real-world DENV population dynamics.

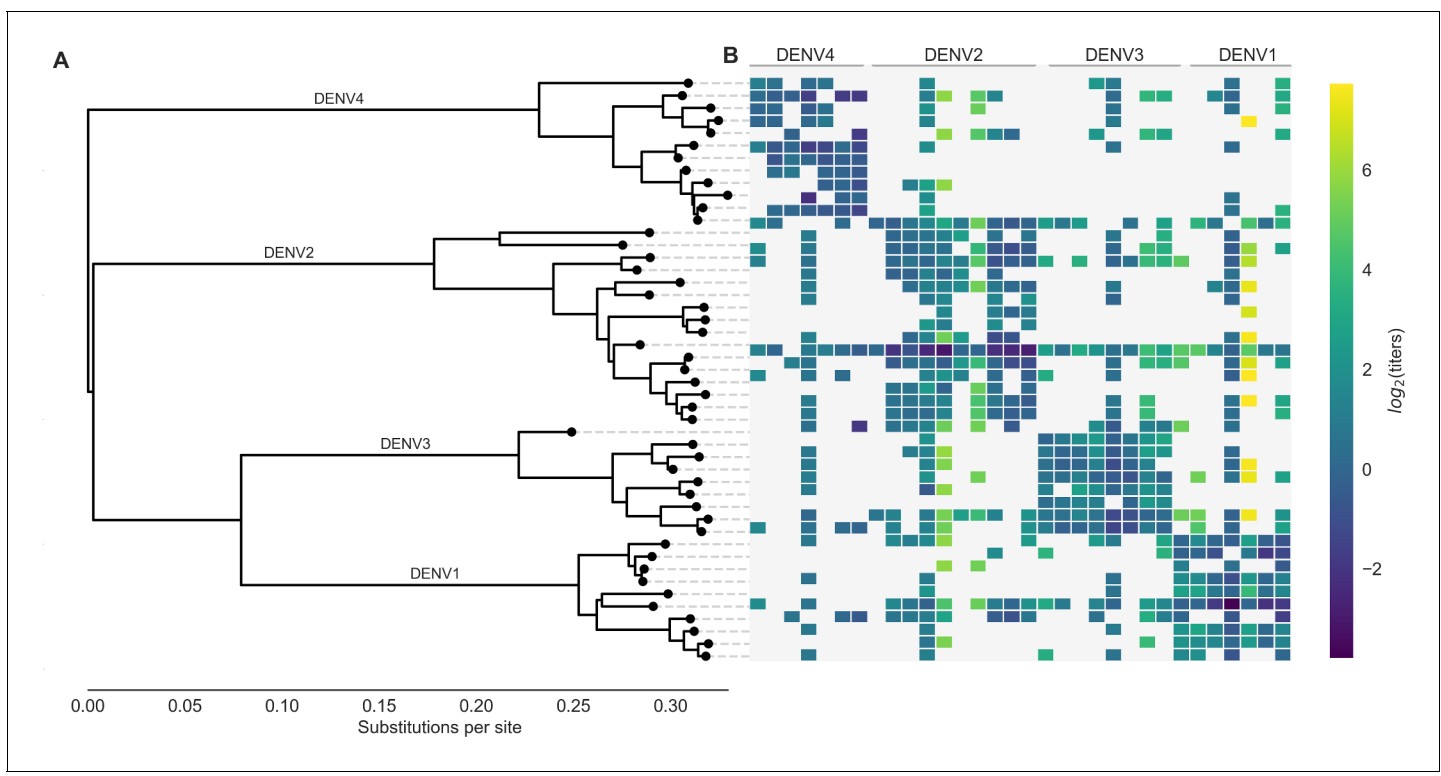

**Figure 1.** Phylogeny of dengue virus sequences and normalized antigenic distances. (**A**) Maximum likelihood phylogeny of the *E* (envelope) gene from titered dengue viruses. Notably, each of the four serotypes contains substantial genetic diversity. (**B**) Pairwise antigenic distances were estimated by *Katzelnick et al. (2015)* using plaque reduction neutralization titers (PRNT50, see Materials and methods). Aggregated titer values are standardized such that the distance between autologous virus-serum pairs is 0, and each titer unit corresponds to a two-fold change in PRNT50 value. Light gray areas represent missing data. Larger values correspond to greater antigenic distance.

The online version of this article includes the following source data and figure supplement(s) for figure 1:

**Source data 1.**

**Figure supplement 1.** Titer value symmetry.

## Results

### Dengue neutralization titer data

Antigenic distance between a pair of viruses $i$ and $j$ is experimentally quantified using neutralization titers, which measure how well serum drawn after infection with virus $j$ is able to neutralize virus $i$ in vitro (*Russell and Nisalak, 1967*). Throughout the following, we refer to serum raised against virus $j$ as serum $j$ for brevity. To measure the pairwise antigenic distances for a panel of diverse DENV viruses (*Figure 1*), *Katzelnick et al. (2015)* infected naive non-human primates (NHP) with each virus, drew sera at 3 months post-infection, and then titered this sera against a panel of test viruses. To compare patterns of cross-protection in NHP and humans, they also drew sera from 31 study participants 6 weeks after inoculation with a monovalent component of the NIH dengue vaccine candidate. This sera was also titered against a broad panel of DENV viruses. As originally reported, we find generally consistent patterns of neutralization between the NHP and human sera data; see *Katzelnick et al. (2015)* for a detailed comparison. In total, our dataset consists of 454 NHP sera titers spanning the breadth of DENV diversity, and 728 human sera titers providing deep coverage of a small subset of viruses.

To normalize these measurements, we take the $\log_2$ of each value, such that one antigenic unit corresponds to a two-fold drop in neutralization, and we define antigenic distance between autologous serum-virus pairs (i.e. virus $i$ and serum $i$) as zero. Normalized antigenic distance between virus $i$ and serum $j$ is thus calculated as $D_{ij} = \log_2(T_{ii}) - \log_2(T_{ij})$, such that a higher value of $D_{ij}$ indicates that serum $j$ is less effective at neutralizing virus $i$, implying greater antigenic distance between viruses $i$ and $j$. For brevity, these normalized titer values are hereafter referred to simply as $\log_2$(titers).

The full dataset of standardized titer values is shown in *Figure 1B*. Here, we see that homotypic virus-serum pairs are more closely related antigenically than heterotypic pairs. However, we also observe large variance around this trend, both within and between serotypes. This suggests that treating each serotype as antigenically uniform potentially overlooks important antigenic heterogeneity across viruses within each serotype.

### Dengue antigenic evolution corresponds to genetic divergence

Titer measurements are prone to noise, and there is a limited amount of available titer data. If the antigenic heterogeneity observed in the raw data is truly the result of an underlying evolutionary process, we expect that differences in antigenic phenotype correspond to underlying mutations in surface proteins. Dengue has two surface proteins, prM (membrane) and E (envelope). While previous studies have identified epitopes on both prM and E, it is believed that antibodies involved in ADE primarily target prM, while neutralizing antibodies primarily target E (*de Alwis et al., 2014*). The assay used to generate this titer dataset captures neutralization, but does not capture the effects of ADE; we thus focus our analysis on the E gene.

To fully map the relationship between DENV genetic and antigenic evolution, we adapt a substitution-based model originally developed for influenza (*Neher et al., 2016*). Conceptually, this model predicts titer values through three steps. First, we align E gene sequences from titered dengue viruses and catalog the amino acid mutations between each serum strain and test virus strain in our dataset. Next, we infer how much antigenic change is attributable to specific mutations by constructing a parsimonious model that links normalized antigenic distances to observed mutations. This assigns each mutation $m$ an antigenic effect size, $d_m \geq 0$; forward and reverse mutations are assigned separate values of $d_m$. With this in hand, we estimate the asymmetrical antigenic distance $\hat{D}_{ij}$ between all pairs of sera and test viruses by summing over $d_m$ for all mutations observed between the serum and the test virus (Materials and methods, *Equation 2*).

To learn these values of $d_m$, we first split our dataset into training (random 90% of measurements) and test data (the remaining 10% of values). We take the training data and fit $d_m$ for each mutation that is observed two or more times, subject to regularization as follows (also detailed in Materials and methods, *Equation 3*). Parsimoniously, we expect that antigenic change is more likely to be incurred by a few key mutations than by many mutations; correspondingly, our prior expectation of values of $d_m$ is exponentially distributed such that most values of $d_m = 0$. This is directly analogous to lasso regression to identify a few parameters with positive weights and set other parameters

to 0 (*Tibshirani, 1996*). Additionally, some viruses have greater binding avidity, and some sera are more potent than others (*Figure 1—figure supplement 1*); these 'row' and 'column' effects, respectively, are normally distributed and are taken into account when training the model. The model uses convex optimization to learn the values of $d_m$ that minimize the sum of squared errors (SSE) between observed and predicted titers in the training data. We thus learn model parameters from the training data, and then use those parameters to predict test data values. We assess model performance by comparing the predicted test titer values to the actual values, aggregated across 100-fold Monte Carlo cross validation.

This model formulation is an effective tool for estimating antigenic relationships between viruses based on their genetic sequences. On average across cross-validation replicates, this model yields a root mean squared error (RMSE) of 0.75 when predicting titers relative to their true value (95% CI 0.74–0.77, RMSE), and explains 78% of the observed variation in neutralization titers overall (95% CI 0.77–0.79, Pearson $R^2$). This is comparable to the model error from a cartography-based characterization of the same dataset (RMSE 0.65–0.8 $\log_2$ titer units) (*Katzelnick et al., 2015*). Prediction error was comparable between human and non-human primate sera, indicating that these genetic determinants of antigenic phenotypes are not host species-specific (*Figure 2—figure supplement 2*).

The 48 strains included in the titer dataset (as serum strains, test virus strains, or both) are 25.7% divergent on average (amino acid differences in *E*). Pairwise comparisons of all serum strains and test viruses yields 1534 unique mutations that are observed at least twice. Our parsimonious model attributes antigenic change to a total of 49 specific mutations and four colinear mutation clusters (each consisting of 2–6 co-occurring mutations) (*Figure 2*, *Table 1*). Each of these mutations confers 0.01–2.11 (median 0.19) $\log_2$ titer units of antigenic change; 27 mutations or mutation clusters have $d_m \geq 0.2$. These mutations span all domains of *E*, and most occur both between and within serotypes (*Figure 2*).

## Each serotype of dengue contains moderate antigenic heterogeneity

By linking antigenic change to specific mutations, we are able to estimate unmeasured antigenic distances between any pair of viruses in the dataset based on their genetic differences. As an example, we estimated the antigenic distance between serum raised against each monovalent component of the NIH vaccine candidate and all other viruses in the dataset. As shown in *Figure 3*, vaccine-elicited antibodies result in strong homotypic neutralization, but heterotypic cross-neutralization varies widely between specific strains. This has important ramifications for vaccine design and trial evaluation.

We also observe antigenic heterogeneity at the genotype level. On average, heterotypic genotypes are separated by 6.9 antigenic mutations (or colinear mutation clusters) and 2.18 $\log_2$ titers. Homotypic genotypes are separated by a mean of 1.9 antigenic mutations, conferring a total of 0.30 $\log_2$ titers of antigenic distance (*Figure 4*). Notably, the titer dataset spans the breadth of canonical DENV genotypes, but in most cases lacks the resolution to detect within-genotype antigenic diversity. We thus expect that these results represent a lower bound on the true extent of DENV intraserotype antigenic diversity.

In summary, we have identified a small number of antigenically relevant mutations that explain most of the observed antigenic heterogeneity in dengue, as indicated by neutralization titers. These mutations occur both between and within serotypes, suggesting that dengue antigenic evolution is an ongoing, although gradual, process. This results in strain-specific and genotype-level antigenic variation, although the scale of this variation is small compared to serotype-level differences. From this, we conclude that there is antigenic variation within each serotype of DENV, and that this is driven by underlying genetic divergence.

## Antigenic novelty predicts serotype success

From the titer model, we find evidence that homotypic genotypes of DENV vary in their ability to escape antibody neutralization. However, antibody neutralization is only one of many factors that shape epidemic patterns. We investigate whether the observed antigenic diversity influences dengue population dynamics in the real world.

The size of the viral population (i.e. prevalence, commonly analyzed using SIR models as reviewed in *Lourenço et al., 2018*) is determined by many complex factors, and reliable values for population

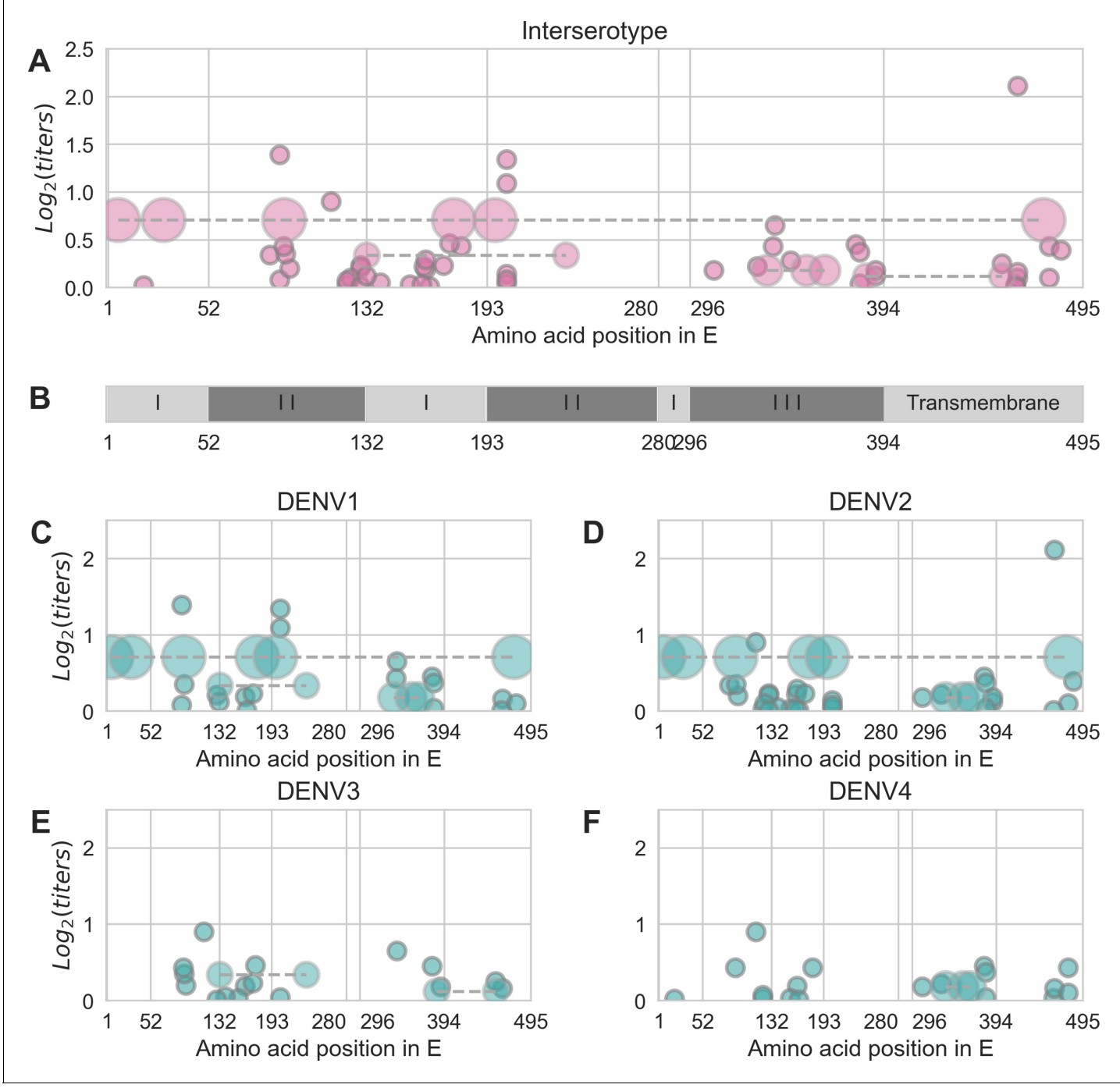

**Figure 2.** Distribution and effect size of antigenic mutations. Each point represents one antigenically relevant mutation or colinear mutation cluster. Clustered mutations are connected with dashed lines with point size proportionate to cluster size (N = 2–6). The x axis indicates mutations' position in *E*, relative to each functional domain as noted in (B). The y axis indicates antigenic effect size.

The online version of this article includes the following source data and figure supplement(s) for figure 2:

**Source data 1.**

**Figure supplement 1.** Genotype as site E 390 across dengue phylogeny.

**Figure supplement 2.** Titer prediction error by serum strain and species.

**Table 1.** Antigenically relevant mutations.
Each entry represents a mutation (or colinear cluster of mutations) inferred by the titer model to have a non-zero antigenic effect size $d_m$ (shown in parentheses).

| | | |
|---|---|---|
| I6V, S29G, F90Y, T176P, V197I, L475M (0.71) | D154E (0.03) | T339I (0.65) |
| A19T (0.02) | K160V (0.03) | V347A (0.28) |
| N83K (0.34) | E161T (0.22) | I380V (0.45) |
| A88K (0.08) | A162I (0.19) | V382A (0.04) |
| A88Q (1.39) | I162A (0.29) | V382I (0.37) |
| Y90F (0.43) | I164V (0.01) | N385K, V454I (0.12) |
| V91I (0.35) | T171S (0.23) | D390S (0.12) |
| K93R (0.20) | V174E (0.46) | N390S (0.18) |
| V114I (0.90) | E180T (0.43) | V454T (0.25) |
| L122S (0.03) | N203E (0.04) | V461F (0.01) |
| S122L (0.07) | N203K (0.08) | F461I (0.03) |
| N124K (0.10) | D203N (0.14) | I462V (0.10) |
| V129I (0.01) | E203N (1.09) | L462I (0.16) |
| I129V (0.21) | E203D (1.34) | V462L (2.11) |
| I129A (0.23) | I308V (0.18) | T478S (0.10) |
| Y132I (0.12) | G330D (0.22) | S478M (0.43) |
| Y132P, R233Q (0.34) | I335V, N355T, P364V(0.18) | V484I (0.39) |
| I139V (0.05) | L338E (0.43) | |

prevalence are largely unavailable. Contrastingly, the composition of the viral population (i.e. the relative frequency of each viral clade currently circulating) can be estimated over time by examining historical sequence data (*Lee et al., 2018*; *Neher et al., 2016*), and is primarily driven by viral fitness (*Bedford et al., 2011*).

In meaningfully antigenically diverse viral populations, antigenic novelty (relative to standing population immunity) contributes to viral fitness: as a given virus *i* circulates in a population, the proportion of the population that is susceptible to infection with *i*–and other viruses antigenically similar to *i*–decreases over time as more people acquire immunity (*Bedford et al., 2012*; *Luksza and Lässig, 2014*). Antigenically novel viruses that are able to escape this population immunity are better able to infect hosts and sustain transmission chains, making them fitter than the previously circulating viruses (*Zhang et al., 2005*; *Bedford et al., 2012*; *Gupta et al., 1998*; *Wearing and Rohani, 2006*; *Lourenço and Recker, 2013*). Thus, if antigenic novelty constitutes a fitness advantage for DENV, then we would expect greater antigenic distance from recently circulating viruses to correlate with higher growth rates.

To test this hypothesis, we examine the composition of the dengue virus population in Southeast Asia from 1970 to 2015. We estimate the relative population frequency of each DENV serotype at three month intervals, $x_i(t)$ (*Figure 5A*), based on their observed relative abundance in the 'slice' of the phylogeny corresponding to each timepoint (N = 8644 viruses; see Materials and methods, *Equations 4-5*). While there is insufficient data to directly compare these estimated frequencies to regional case counts, we see good qualitative concordance between frequencies similarly estimated for Thailand and previously reported case counts from Bangkok (*Figure 5—figure supplement 1*).

Fitter virus clades increase in frequency over time, such that $x_i(t + dt) > x_i(t)$. It follows that these clades have a growth rate—defined as the fold-change in frequency over time—greater than one: $\frac{x_i(t+dt)}{x_i(t)} > 1$. To isolate the extent to which antigenic fitness contributes to clade success and decline, we extend work by *Luksza and Lässig (2014)* to build a simple model that attempts to predict clade growth rates based on two variables: the antigenic fitness of the clade at time $t$, and a time-invariant-free parameter representing the intrinsic fitness of the serotype the clade belongs to. We estimate the antigenic fitness of clade *i* at time *t* as a function of its antigenic distance from each viral

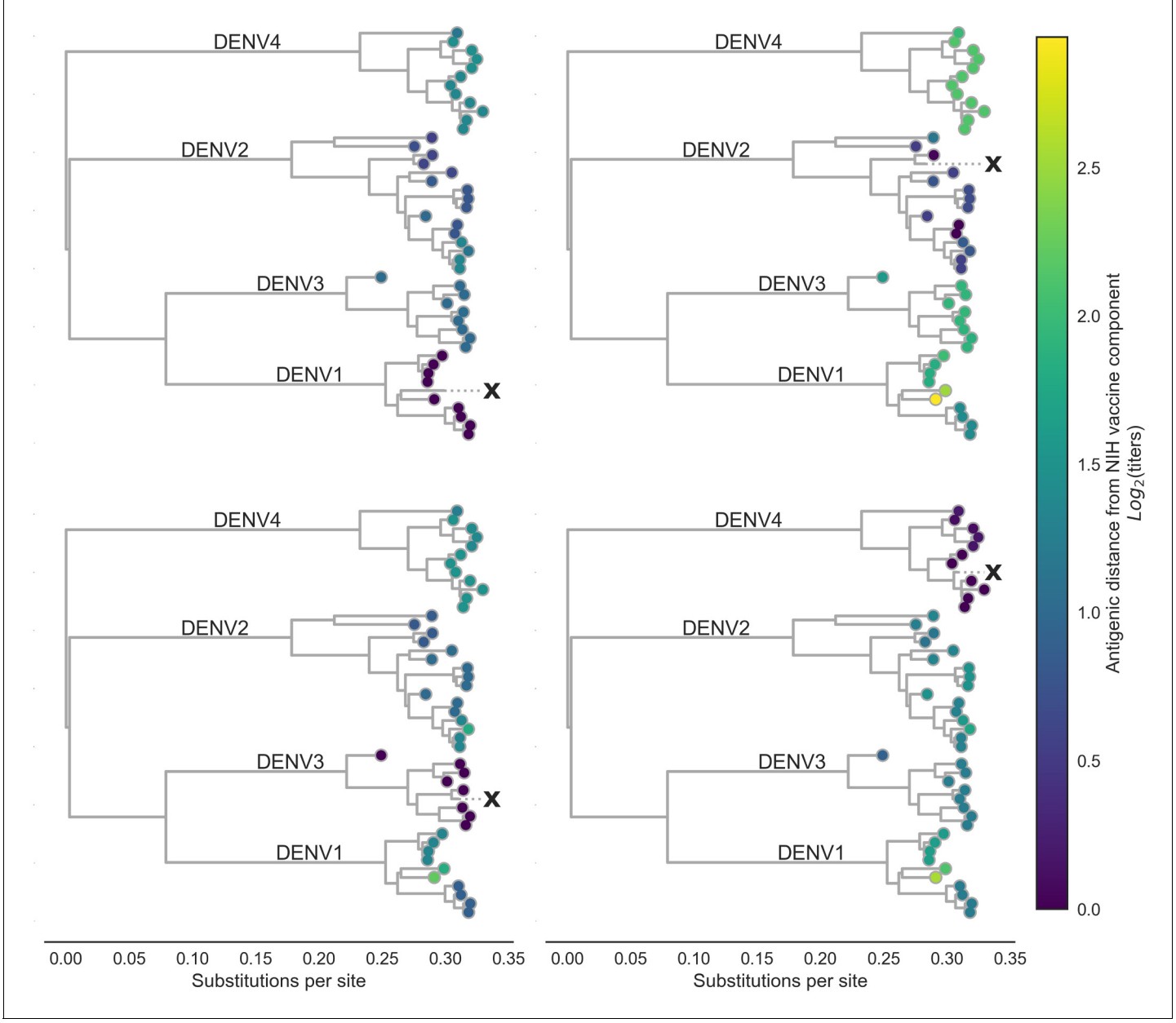

**Figure 3.** Antigenic distance from NIH vaccine strains. By assigning a discrete increment of antigenic change to each mutation, we can estimate the asymmetrical antigenic distance between any serum strain and test virus strain based on their genetic differences. Here, we show the estimated antigenic distance between serum raised against each monovalent component of the NIH vaccine candidate (indicated as 'X') and each test virus in the tree.

clade *j* that has circulated in the same population over the previous 2 years, weighted by the relative frequency of *j* and adjusted for waning population immunity (***Figure 5B***; Materials and methods, ***Equation 6-10***). Growth rates are estimated based on a 2-year sliding window (***Figure 5C***).

This simple model explains 54.7% of the observed variation in serotype growth rates, and predicts serotype growth vs. decline correctly for 66.0% of predictions (***Figure 5D***). This suggests that antigenic fitness is a major driver of serotype population dynamics. This also demonstrates that this model captures key components of dengue population dynamics; examining the formulation of this model in more detail can yield insights into how antigenic relationships influence DENV population composition. The fitness model includes six free parameters that are optimized such that the model most accurately reproduces the observed fluctuations in DENV population composition (minimizing

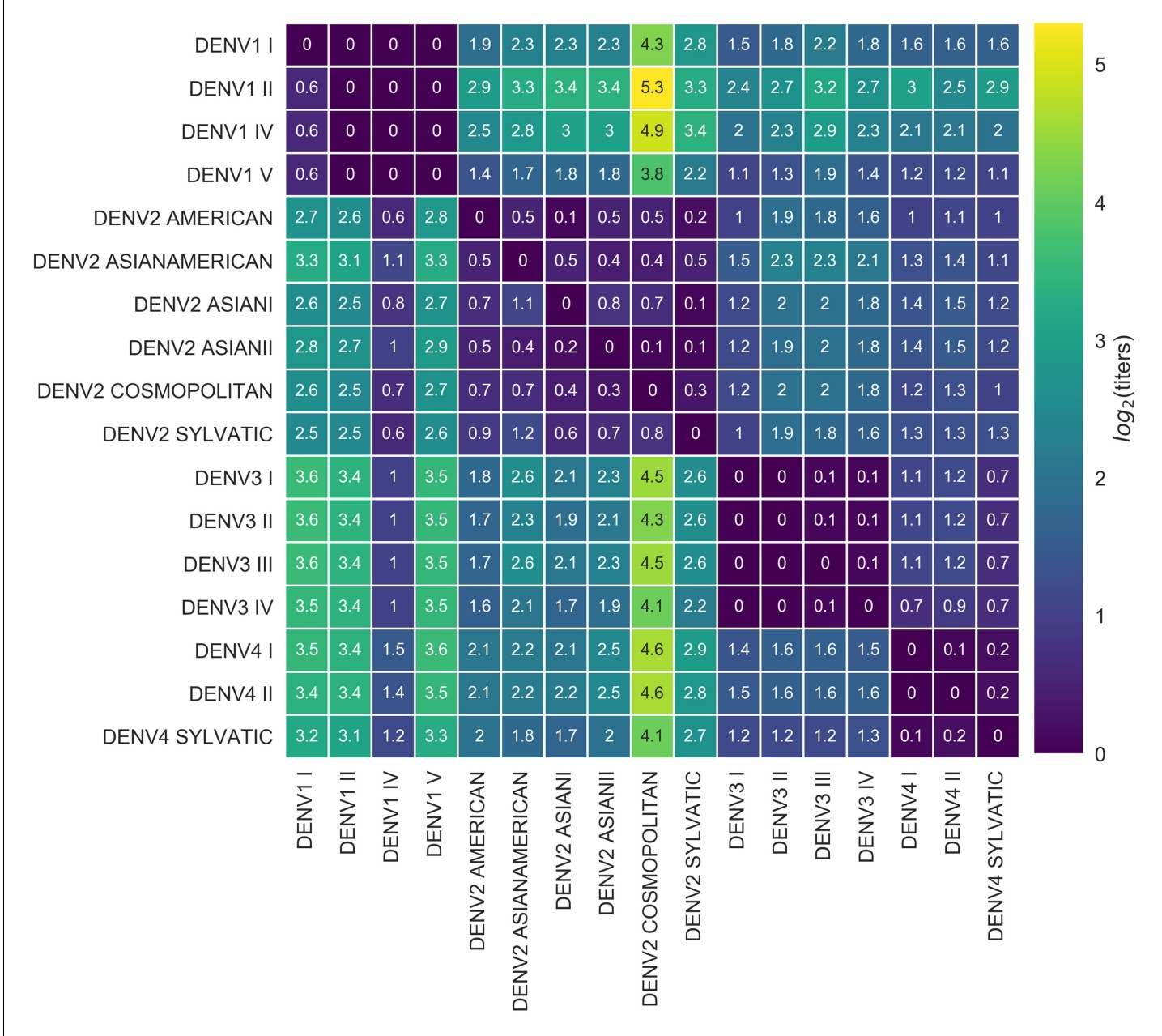

**Figure 4.** Titer distance by genotype. Values represent the mean interpolated antigenic distance between canonical dengue genotypes (in standardized $\log_2$ titer units). Columns represent sera; rows represent test viruses.

The online version of this article includes the following source data for figure 4:

**Source data 1.**

the RMSE of frequency predictions, see Materials and methods). We find that serotype fluctuations are consistent with a model wherein population immunity wanes linearly over time, with the probability of protection dropping by about 63% per year for the first 2 years after primary infection. This model assumes no fundamental difference between homotypic and heterotypic reinfection; rather, homotypic immunity is assumed to wane at the same rate as heterotypic immunity, but starts from a higher baseline of protection based on closer antigenic distances. We also find that these dynamics are best explained by intrinsic fitness that moderately varies by serotype (*Table 2*); we are not aware of any literature that directly addresses this observation via competition experiments. However,

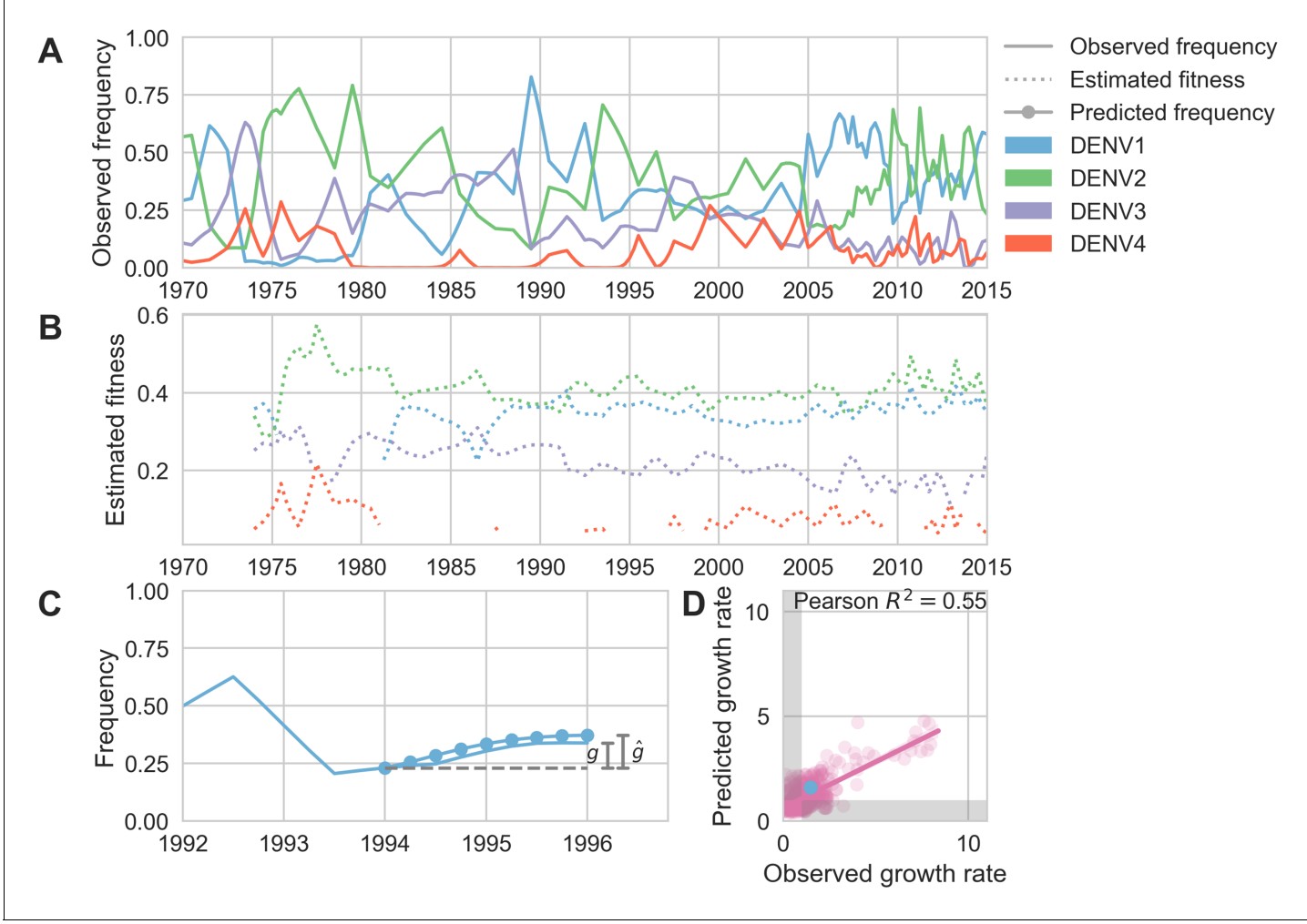

**Figure 5.** Antigenic novelty predicts serotype success. (**A**) The relative frequency of each serotype, $x_i$, in Southeast Asia estimated every 3 months based on available sequence data. (**B**) Total fitness of each serotype. We calculate antigenic fitness for each serotype over time as its frequency-weighted antigenic distance from recently circulating viruses. We then add this to a time-invariant intrinsic fitness value to calculate total fitness. (**C**) DENV1 frequencies between 1994 and 1996 alongside model projection. At each timepoint $t$, we blind the model to all empirical data from timepoints later than $t$ and predict each serotype's future trajectory based on its initial frequency, time-invariant intrinsic fitness, and antigenic fitness at time $t$ (Materials and methods, *Equation 11*). We predict forward in 3-month increments for a total prediction period of $dt = 2$ years. At each increment, we use the current predicted frequency to adjust our estimates of antigenic fitness on a rolling basis (Materials and methods, *Equation 15*). (**D**) Predicted growth rates, $\hat{g} = \frac{\hat{x}_i(t+dt)}{x_i(t)}$, compared to empirically observed growth rates, $g = \frac{x_i(t+dt)}{x_i(t)}$. Predicted and empirical growth rate of the example illustrated in (**C**) is shown in (**D**) as the blue point. Serotype growth versus decline is accurate (i.e. the predicted and actual growth rates are both >1 or both <1, all points outside the gray area) for 66% of predictions.

The online version of this article includes the following source data and figure supplement(s) for figure 5:

**Source data 1.**
**Figure supplement 1.** Case counts versus clade frequencies in Thailand.
**Figure supplement 2.** Simulated serotype frequencies (model parameters).
**Figure supplement 3.** Simulated serotype frequencies.

intrinsic fitness alone is unable to predict serotype dynamics (*Table 3*) and relative strength of antigenic fitness and intrinsic fitness are approximately matched in determining overall serotype fitness.

## Antigenic novelty also partially predicts genotype success

To estimate how well antigenic fitness predicts genotype dynamics, we used the same model to predict genotype success and decline. As before, fitness of genotype $i$ is based on the intrinsic fitness

**Table 2.** Optimized fitness model parameters for primary analysis.

| Parameter | Value | Description |
|---|---|---|
| $\beta$ | 1.02 | Slope of linear relationship between population immunity and viral fitness |
| $\gamma$ | 0.83 | Proportion of titers waning each year since primary infection |
| $\sigma$ | 0.76 | Slope of linear relationship between titers and probability of protection |
| $f_0^{(1)}$ | 0.74 | Relative intrinsic fitness of DENV1 |
| $f_0^{(2)}$ | 0.84 | Relative intrinsic fitness of DENV2 |
| $f_0^{(3)}$ | 0.50 | Relative intrinsic fitness of DENV3 |
| $f_0^{(4)}$ | 0.00 | Relative intrinsic fitness of DENV4 (fixed) |

of the serotype $i$ belongs to, and the antigenic distance between $i$ and each other genotype, $j$, that has recently circulated (*Equation 6-10*). For genotypes, we can calculate antigenic distance between $i$ and $j$ at either the serotype level or the genotype level. In the 'interserotype model', we treat each serotype as antigenically uniform, and assign the mean serotype-level antigenic distances to all pairs of constituent genotypes. In the 'intergenotype model', we incorporate the observed within-serotype heterogeneity, and use the mean genotype-level antigenic distances (as shown in *Figure 4*). If within-serotype antigenic heterogeneity contributes to genotype fitness, then we would expect estimates of antigenic fitness based on the 'intergenotype model' to better predict genotype growth rates.

We find that antigenic fitness contributes to genotype turnover, although it explains less of the observed variation than for serotypes. As for serotypes, intrinsic fitness alone was unable to predict genotype turnover (*Table 3*). When antigenic distance is estimated from the 'interserotype model', we find that our model of antigenic fitness explains approximately 28.6% of the observed variation in genotype growth rates, and correctly predicts genotype growth vs. decline 66.6% of the time (*Figure 6C*). Perhaps surprisingly, more precise estimates of antigenic distance between genotypes from the 'intergenotype model' does not improve our predictions of genotype success ($R^2 = 0.254$, 61.0% accuracy; *Figure 6D*, *Table 3*). This suggests that although we find strong evidence that

**Table 3.** Fitness model performance comparisons.
Here we compare the performance of the antigenically-informed fitness models to model performance under two null formulations. In the 'equal' null model, all clades are assigned equal fitness (i.e. antigenic and intrinsic fitness are set to 0, Equation 17; Equation 18). In the 'intrinsic' null model formulation, only the serotype-specific, time-invariant intrinsic fitness values contribute to clade fitness (i.e. antigenic fitness is set to 0, Equation 19; Equation 20). For both formulations of generalized waning, all other parameters were set to the values reported in *Table 2* (optimized for RMSE). Parameters for heterotypic waning were optimized separately.

| Resolution | Fitness model | Waning | RMSE | Pearson $R^2$ | Accuracy |
|---|---|---|---|---|---|
| Serotype | Interserotype | Generalized | 0.105 | 0.547 | 0.660 |
| Serotype | Equal fitness null | Generalized | 0.130 | 0.000 | 0.480 |
| Serotype | Intrinsic fitness null | Generalized | 0.140 | 0.042 | 0.510 |
| Genotype | Interserotype | Generalized | 0.062 | 0.286 | 0.666 |
| Genotype | Intergenotype | Generalized | 0.062 | 0.254 | 0.610 |
| Genotype | Equal fitness null | Generalized | 0.070 | 0.000 | 0.440 |
| Genotype | Intrinsic fitness null | Generalized | 0.072 | 0.032 | 0.530 |
| Serotype | Interserotype | Heterotypic | 0.109 | 0.533 | 0.666 |
| Genotype | Interserotype | Heterotypic | 0.063 | 0.291 | 0.661 |
| Genotype | Intergenotype | Heterotypic | 0.063 | 0.203 | 0.599 |

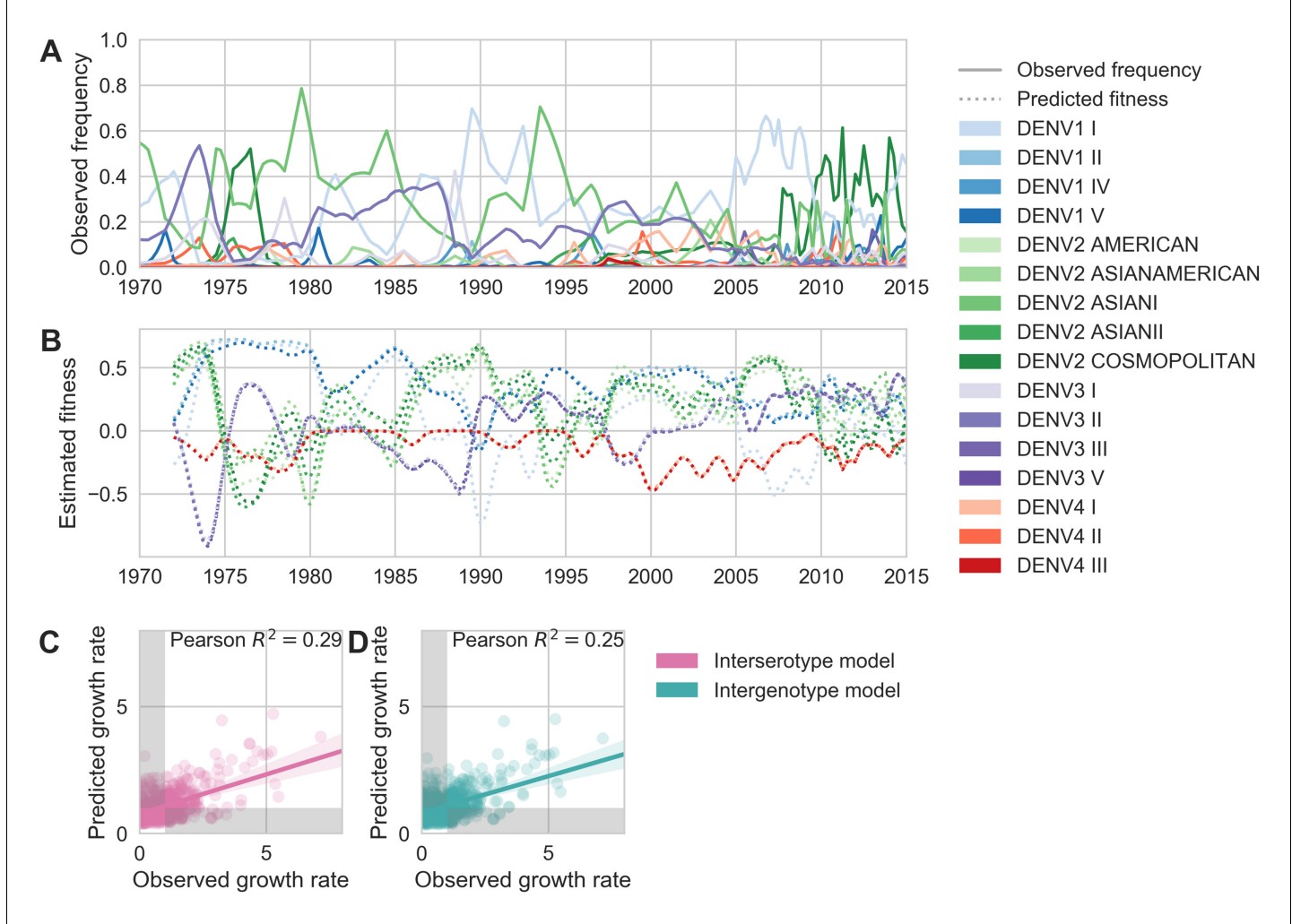

**Figure 6.** Antigenic novelty partially predicts genotype success. (**A**) Relative frequencies of each canonical dengue genotype across Southeast Asia, estimated from available sequence data. (**B**) Antigenic fitness is calculated for each genotype as its frequency-weighted antigenic distance from recently circulating genotypes. We then add this to a time-invariant, serotype-specific intrinsic fitness value to calculate total fitness (shown here, arbitrary units). We assess antigenic distance at either the 'intergenotype' or the 'interserotype' resolution. In this panel, we show total fitness over time, incorporating estimates of antigenic fitness derived from the 'intergenotype' model. (**C, D**) Fitness estimates were used to predict clade growth rates over 2 years, compounding immunity every 3 months based on predicted frequency changes (Materials and methods *Equation 15*). Here, we compare observed vs. predicted growth rates for both formulations of the fitness model (using fitness derived from either 'interserotype' or 'intergenotype' antigenic distances). Growth versus decline was accurate (predicted and actual growth rates both >1 or both <1, points outside the gray shaded area) for 67% and 61% of predictions, respectively.

The online version of this article includes the following source data for figure 6:

**Source data 1.**

---

genotypes vary in their ability to escape neutralizing antibodies, these differences are subtle enough that they do not impact broad-scale regional dynamics over time.

## Discussion

### Within-serotype antigenic heterogeneity

We show that mapping antigenic change to specific mutations and interpolating across the DENV alignment is able to explain a large majority of the observed variation in antigenic phenotypes, as measured by neutralization titers. We identify 49 specific mutations and 4 colinear mutation clusters that contribute to antigenic variation, of which 27 mutations or mutation clusters have an antigenic

impact of 0.20 $\log_2$ titers or greater. These mutations span all major domains of E, and occur both within and between serotypes. This demonstrates that DENV antigenic divergence is closely coupled to genetic divergence. We use these mutations to infer unmeasured antigenic relationships between viruses, revealing substantial within-serotype antigenic variation. For comparison, we reconstructed the ancestral sequence of each serotype and constrained the model to only permit antigenic change to be attributed to these serotype-level differences. While this interserotype-only model predicts titers to a reasonable degree, we find that it has higher error (RMSE = 0.86) than the full model which accounts for within-serotype heterogeneity (RMSE = 0.79; *Table 4*). This supports and expands upon previous reports (*Katzelnick et al., 2015*; *Forshey et al., 2016*; *Waggoner et al., 2016*) that the null hypothesis of antigenically uniform serotypes is inconsistent with observed patterns of cross-protection and susceptibility.

Consistent with the relatively long timescale of dengue evolution, we observe many sites in the dengue phylogeny to have mutated multiple times. These represent instances of parallelism, reversion and homoplasy. For example, we observe that site 390 is consistently S in DENV1, N in DENV3 and H in DENV4, while DENV2 genotypes show a mixture of D, N and S (*Figure 2—figure supplement 1*). We estimate an antigenic impact of 0.18 $\log_2$ titers of the N390S mutation. Our model predicts that the parallel N390S mutations in DENV1 and DENV2 Cosmopolitan makes these viruses slightly more antigenically similar rather than more antigenically distinct. Along these lines, we compared the 'substitution' model to a similar model formulation (termed the 'tree' model) which assigns $d_m$ values to individual branches in the phylogeny, rather than to individual mutations, so that each branch with a positive $d_m$ value increases antigenic distance between strains (*Neher et al., 2016*). As expected from the high degree of homoplasy across the dengue phylogeny, we observe that the 'substitution' model outperforms the 'tree' model in predicting titers in validation datasets (*Table 3*).

To investigate the impact of this observed variation, we examine patterns of neutralization in response to vaccination with each monovalent component of the NIH vaccine candidate (*Figure 3*). Here, we see that each monovalent component elicits broad homotypic protection, but levels of heterotypic cross protection vary widely between heterotypic genotypes. This is consistent with previous reports of genotype-specific interactions between standing population immunity and

**Table 4.** Titer model performance comparisons.
We compared performance across several different variations of the titer model. As described in *Neher et al. (2016)*, incremental antigenic change can be assigned to either amino acid substitutions ('Substitution' model) or to branches in the phylogeny ('Tree' model). For each of these models, we can constrain the model such that antigenic change is allowed to occur only between serotypes ('Interserotype') or between AND within serotypes ('Full'). For the substitution model, we constrain the interserotype model by reconstructing the amino acid sequence of the most recent common ancestor for each serotype and allowing the model to assign antigenic change only to mutations between these ancestral sequences. For the tree model, we constrain the interserotype model by allowing the model to assign antigenic change only to branches in the phylogeny that lie between serotypes. We also assess the impact of the virus avidity and serum potency terms, $v_a$ and $p_b$. For all models and metrics, we report the mean and 95% confidence interval across 100-fold Monte Carlo cross validation with random 90%:10%, training:test splits.

| Model | Antigenic resolution | $v_a$ And $p_b$ | RMSE | Pearson $R^2$ |
|---|---|---|---|---|
| Substitution | Full | Yes | 0.75 (0.74–0.77) | 0.78 (0.77–0.79) |
| Substitution | Full | No | 1.13 (1.11–1.16) | 0.50 (0.48–0.52) |
| Substitution | Interserotype | Yes | 0.86 (0.85–0.88) | 0.72 (0.70–0.73) |
| Substitution | Interserotype | No | 0.86 (0.84–0.87) | 0.71 (0.70–0.72) |
| Tree | Full | Yes | 0.84 (0.83–0.86) | 0.72 (0.71–0.73) |
| Tree | Full | No | 1.40 (1.38–1.42) | 0.24 (0.23–0.26) |
| Tree | Interserotype | Yes | 0.87 (0.85–0.88) | 0.70 (0.69–0.71) |
| Tree | Interserotype | No | 0.86 (0.84–0.88) | 0.72 (0.71–0.73) |

subsequent heterotypic epidemics as modulating epidemic severity (*OhAinle et al., 2011*; *Kochel et al., 2002*). We hypothesize that this observed within-serotype variation primarily effects heterotypic secondary infection outcomes, rather than modulating homotypic immunity. Although we note that *Juraska et al. (2018)* demonstrate that vaccine efficacy decreases with increasing amino acid divergence of breakthrough infections from the vaccine insert.

Overall, we expect that these antigenic phenotypes represent a lower bound on the extent, magnitude, and nature of antigenic heterogeneity with DENV. Our current titer dataset spans the breadth of DENV diversity, but due to small sample size, it lacks the resolution to detect most sub-genotype antigenic variation. The appearance of the deep antigenic divergence of the four serotypes, and the more recent antigenic divergences within each serotype, suggest that DENV antigenic evolution is likely an ongoing, although gradual, process. We therefore expect that future studies with richer datasets will find additional antigenic variation within each genotype. This dataset also contains many left-censored titer values, where we know two viruses are at least $T$ titer units apart, but do not know exactly how far apart. If we knew the true value of these censored titers, many of them would indicate larger antigenic distances than the reported values, $T$, which are used to train the model. Thus, it is likely that our model systematically underestimates the magnitude of titer distances.

Finally, antibody neutralization and escape (as measured by PRNT titers) is only one component of the immune response to DENV. Although analysis of a longitudinal cohort study shows that these neutralization titers correlate with protection from severe secondary infection, it is unclear how PRNT titers correspond to antibody-dependent enhancement (*Katzelnick et al., 2016*). It is also important to note that DENV case outcomes are partially mediated by interactions with innate and T-cell immunity, the effects of which are not captured in neutralization titers (*Green et al., 2014*). Overall, while richer datasets and the development of more holistic assays will be required in order to fully characterize the extent of DENV antigenic diversity, it is clear that the four-serotype model is insufficient to explain DENV antigenic evolution.

## Viral clade dynamics

We use these inferred antigenic relationships to directly quantify the impact of antigenic fitness on DENV population composition. To do so, we measure serotype frequencies across Southeast Asia over time and construct a model to estimate how they will fluctuate (Materials and methods, *Equation 6-16*). This model places a fitness value on each serotype that derives from a constant intrinsic component alongside a time-dependent antigenic component. Antigenic fitness declines with population immunity, which is accumulated via the recent circulation of antigenically similar viruses. Our primary model parameterization assumes that both heterotypic and homotypic immunity wane linearly over time at the same rate, with homotypic immunity starting from a higher baseline of protection based on closer antigenic distances. We compared this to a secondary model parameterization with only heterotypic waning (see Materials and methods), under which we observe similar model performance (*Table 3*).

We find that antigenic fitness is able to explain much of the observed variation in serotype growth and decline (*Figure 5*). Forward simulations under the optimized parameter set display damped oscillations around the serotype-specific 'set points' determined by intrinsic fitnesses, but intrinsic fitness alone is unable to explain serotype fluctuations ($R^2 = 0.04$; *Table 3*, *Figure 5—figure supplement 2*). This demonstrates that although intrinsic fitness plays an important role in dictating long-term dynamics, wherein particular serotypes tend to circulate at low frequency (e.g. DENV4) and others at high frequency (e.g. DENV1 and DENV2), antigenic fitness plays out on shorter-term time scales, dictating circulation over several subsequent years.

We similarly use this model to quantify the effect of within-serotype antigenic variation on the success and decline of canonical DENV genotypes (*Figure 6*). As above, genotype antigenic fitness declines with population immunity. Here, we estimate population immunity based on antigenic distance from recently circulating genotypes, using distances that are either genotype-specific or based only on the serotype that each genotype belongs to. We then directly compare how strongly these coarser serotype-level versus specific genotype-level antigenic relationships impact DENV population dynamics. Overall, we find that antigenic fitness explains a moderate portion of the observed variation in genotype growth and decline. Surprisingly, however, we find that incorporating within-

serotype antigenic differences does not improve our predictions (*Figure 6C–D*). We suggest two possible explanations for this observation.

First, it may be that although genotypes are antigenically diverse, these differences do not influence large-scale regional dynamics over time. We may then hypothesize that within-serotype antigenic heterogeneity mediates disease severity, but does not influence infection or onward transmission. This hypothesis is consistent with the findings of *Nagao and Koelle (2008)*, who demonstrated that dengue epidemiological dynamics are compatible with a model wherein immunity confers protection against severe symptoms, but not asymptomatic infection. This is also consistent with *Ten Bosch et al. (2018)*'s findings that asymptomatic dengue infections contribute to onward transmission.

Alternatively, this lack of signal could be methodologically explained if either (A) genotype-level frequency trajectories estimated from public data are overly noisy for this application or (B) our model of antigenic fitness based on PRNT assay data does not match reality, due to either PRNT assay data not well reflecting human immunity or due to our particular model formulation that parameterizes immunity from titer distances (*Equation 6-10*). In the present analysis, we are not able to firmly resolve these disparate possibilities.

These observations are also subject to caveats imposed by the available data and model assumptions. Our estimates of antigenic fitness are informed by the antigenic distances inferred by the substitution model; thus, as above, we are unable to account for nuanced antigenic differences between sub-genotype clades of DENV due to limited titer data. We estimate DENV population composition over time based on available sequence data, pooled across all of Southeast Asia (Materials and methods, *Equation 4*). As the vast majority of cases of DENV are asymptomatic, sequenced viruses likely represent a biased sample of more severe cases from urban centers where patients are more likely to seek and access care. We also assume that Southeast Asia represents a closed viral population with homogeneous mixing. However, increasing globalization likely results in some amount of viral importation that is not accounted for in this model (*Allicock et al., 2012*). Finally, although Southeast Asia experiences hyperendemic DENV circulation, the majority of DENV transmissions are hyper-local (*Salje et al., 2017*), and viral populations across this broad region may not mix homogeneously each season. Thus, it is possible that these sub-serotype antigenic differences impact finer-scale population dynamics, but we lack the requisite data to examine this hypothesis.

## Conclusions

We find that within-serotype antigenic evolution helps explain observed patterns of cross-neutralization among dengue genotypes. We also find that serotype-level population immunity is a strong determinant of viral clade dynamics across Southeast Asia. As richer datasets become available, future studies that similarly combine viral genomics, functional antigenic characterization, and population modeling have great potential to improve our understanding of how DENV evolves antigenically and moves through populations.

## Model sharing and extensions

We have provided all source code, configuration files and datasets at https://github.com/blab/dengue-antigenic-dynamics (copy archived at https://github.com/elifesciences-publications/dengue-antigenic-dynamics), and wholeheartedly encourage other groups to adapt and extend this framework for further investigation of DENV antigenic evolution and population dynamics.

## Materials and methods

### Data

#### Titers

Antigenic distance between pairs of viruses *i* and *j* is experimentally measured using a neutralization titer, which measures how well serum drawn after infection with virus *i* is able to neutralize virus *j* in vitro (*Russell and Nisalak, 1967*). Briefly, two-fold serial dilutions of serum *i* are incubated with a fixed concentration of virus *j*. Titers represent the lowest serum concentration able to neutralize 50% of virus, and are reported as the inverse dilution. We used two publicly available plaque reduction

neutralization titer (PRNT50) datasets generated by *Katzelnick et al. (2015)*. The primary dataset was generated by infecting each of 36 non-human primates with a unique strain of DENV. NHP sera was drawn after 12 weeks and titered against the panel of DENV viruses. The secondary dataset was generated by vaccinating 31 human trial participants with a monovalent component of the NIH DENV vaccine. Sera was drawn after 6 weeks and titered against the same panel of DENV viruses. As discussed in Katzelnick et al., these two datasets show similar patterns of antigenic relationships between DENV viruses. In total, our dataset includes 1182 measurements across 48 virus strains: 36 of these were used to generate serum, and 47 were used as test viruses.

## Sequences

For the titer model analysis, we used the full sequence of *E* (envelope) from the 48 strains in the titer dataset.

For the clade frequencies analysis, we downloaded all dengue genome sequences available from the Los Alamos National Lab Hemorrhagic Fever Virus Database as of March 7, 2018, that contained at least the full coding sequence of *E* (envelope) (total N = 12,645) (*Kuiken et al., 2012*). We discarded sequences which were putative recombinants, duplicates, lab strains, or which lacked an annotated sampling location and/or sampling date. We selected all remaining virus strains that were annotated as a Southeast Asian isolate (total N = 8,644).

For both datasets, we used the annotated reference dataset from *Pyke et al. (2016)* to assign sequences to canonical genotypes.

## Alignments and trees

We used MAFFT v7.305b to align nucleotide *E* gene sequences for each strain before translating the aligned sequences (no frame-shift indels were present) (*Katoh and Standley, 2013*). All maximum likelihood phylogenies were constructed with IQ-TREE version 1.6.8 and the GTR + I + G15 nucleotide substitution model (*Nguyen et al., 2015*).

## Titer model

We compute standardized antigenic distance between virus $i$ and serum $j$ (denoted $D_{ij}$) from measured titers $T_{ij}$ relative to autologous titers $T_{ii}$, such that

$$D_{ij} = \log_2(T_{ii}) - \log_2(T_{ij}). \tag{1}$$

We then average normalized titers across individuals. To predict unmeasured titers, we employ the 'substitution model' from *Neher et al. (2016)* and implemented in Nextstrain (*Hadfield et al., 2018*), which assumes that antigenic evolution is driven by underlying genetic evolution.

In the substitution model, observed titer drops are mapped to mutations between each serum and test virus strain after correcting for overall virus avidity, $v_i$, and serum potency, $p_j$ ('row' and 'column' effects, respectively), so that

$$D_{ij} \approx \hat{D}_{ij} = \sum_m d_m + v_i + p_j, \tag{2}$$

where $d_m$ is the titer drop assigned to each mutation, $m$, between serum $i$ and virus $j$, and $m$ iterates over mutations. We randomly withhold 10% of titer measurements as a test set. We use the remaining 90% of titer measurements as a training set to learn values for virus avidity, serum potency, and mutation effects. As in *Neher et al. (2016)*, we formulate this as a convex optimization problem and solve for these parameter values to minimize the cost function

$$C = \sum_{i,j} (\hat{D}_{ij} - D_{ij})^2 + \lambda \sum_m d_m + \kappa \sum_i v_i^2 + \delta \sum_j p_j^2. \tag{3}$$

We used $\lambda = 3.0$, $\kappa = 0.6$, and $\delta = 1.2$ to minimize test error. Respectively, these terms represent the squared training error; an L1 regularization term on mutation effects, such that most values of $d_m = 0$; and L2 regularization terms on virus avidities and serum potencies, such that they are normally distributed. These parameter values are then used to predict the antigenic distance between all pairs of viruses, $i$ and $j$. We assess performance by comparing predicted to known titer values in

our test data set, and present test error (aggregated from 100-fold Monte Carlo cross-validation) throughout the manuscript.

## Viral clade dynamics

### Empirical clade frequencies

As discussed in *Neher et al. (2016)* and *Lee et al. (2018)*, we estimate empirical clade frequencies from 1970 to 2015 based on observed relative abundances of each clade in the 'slice' of the phylogeny corresponding to each quarterly timepoint.

Briefly, the frequency trajectory of each clade in the phylogeny is modeled according to a Brownian motion diffusion process discretized to 3-month intervals. Relative to a simple Brownian motion, the expectation includes an 'inertia' term that adds velocity to the diffusion and the variance includes a term $x(1-x)$ to scale variance according to frequency following a Wright-Fisher population genetic process. This results in the diffusion process

$$x(t+dt) = \mathcal{N}\left(x(t) + \epsilon\, dx,\ dt\, \sigma^2 x(t)\, (1-x(t))\right) \tag{4}$$

with 'volatility' parameter $\sigma^2$ and inertia parameter $\epsilon$. The term $dx$ is the increment in the previous timestep, so that $dx = x(t) - x(t-dt)$. We used $\epsilon = 0.7$ and $\sigma = 2.0$ to maximize fit to empirical trajectory behavior.

We also include an Bernoulli observation model for clade presence/absence among sampled viruses at timestep $t$. This observation model follows

$$f(x,t) = \prod_{v \in V} x(t) \prod_{v \notin V} (1 - x(t)), \tag{5}$$

where $v \in V$ represents the set of viruses that belong to the clade and $v \notin V$ represents the set of viruses that do not belong to the clade. Each frequency trajectory is estimated by simultaneously maximizing the likelihood of the process model and the likelihood of the observation model via adjusting frequency trajectory $\vec{x} = (x_1, ... x_n)$.

### Population immunity

For antigenically diverse pathogens, antigenic novelty represents a fitness advantage (*Lipsitch and O'Hagan, 2007*). This means that viruses that are antigenically distinct from previously circulating viruses are able to access more susceptible hosts, allowing the antigenically novel lineage to expand. We adapt a simple deterministic model from *Luksza and Lässig (2014)* to directly quantify dengue antigenic novelty and its impact on viral fitness. We quantify population immunity to virus $i$ at time $t$, $P_i(t)$, as a function of which clades have recently circulated in the past $N$ years, and how antigenically similar each of these clades is to virus $i$, so that

$$P_i(t) = \sum_{n=1}^{n=N} \left( w(n) \sum_j \left( x_j(t-n)\, C(D_{ij}) \right) \right), \tag{6}$$

where $D_{ij}$ is the antigenic distance between $i$ and each non-overlapping clade $j$, $n$ is the number of years since exposure, and $x_j(t-n)$ is the relative frequency of $j$ at year $t-n$. Waning immunity is modeled as a non-negative linear function of time following

$$w(n) = \max(1 - \gamma n, 0). \tag{7}$$

The relationship between antigenic distance and the probability of protection, $C$, is also assumed to be non-negative and linear with slope $-\sigma$, such that

$$C(D_{ij}) = \max(1 - \sigma D_{ij}, 0). \tag{8}$$

In addition to this primary analysis, we conducted a secondary analysis with a different parameterization of immunity that removes waning of homotypic immunity while allowing waning of heterotypic immunity. In this case, we assume the relationship between antigenic distance and the probability of protection, $C$, to be 50% at antigenic distance $1/\sigma$ and to wane based on years since infection $n$ modified by $\gamma_{\mathrm{het}}$ following

$$C(D_{ij}, n) = \exp\left(-\sigma\left(1/\gamma_{\text{het}}\right)^n D_{ij}\right). \tag{9}$$

We model the effects of population immunity, $P_i(t)$, on viral antigenic fitness, $f_i(t)$, as

$$f_i(t) = f_0 - \beta P_i(t), \tag{10}$$

where $\beta$ and $f_0$ are fit parameters representing the slope of the linear relationship between immunity and fitness, and the intrinsic relative fitness of each serotype, respectively.

## Frequency predictions

Similar to the model implemented in *Luksza and Lässig (2014)*, we estimate predicted clade frequencies at time $t + dt$ as

$$\hat{x}_i(t + dt) = \frac{x_i(t)e^{f_i(t)dt}}{\sum_i x_i(t)e^{f_i(t)dt}} \tag{11}$$

for short-term predictions (where $dt$<1 year).

We do not attempt to predict future frequencies for clades with $x_i(t)$<0.05.

For long-term predictions, we must account for immunity accrued at each intermediate timepoint between $t$ and $dt$. We divide the interval between $t$ and $dt$ into 3 month timepoints, $[t + u, t + 2u, ..., t + U]$, such that $U = dt$. We then compound immunity based on predicted clade frequencies at each intermediate timepoint following

$$\hat{x}_i(t + u) = x_i(t)e^{f_i(t)u} \tag{12}$$

$$\hat{x}_i(t + 2u) = \hat{x}_i(t + u)e^{f_i(t+u)u} \tag{13}$$

$$...$$

$$\hat{x}_i(t + U) = x_i(t)e^{f_i(t)u}e^{f_i(t+u)u}e^{f_i(t+2u)u}...e^{f_i(t+U)u} \tag{14}$$

$$\hat{x}_i(t + dt) = \hat{x}_i(t + U) = x_i(t)e^{\sum_u f_i(t+u)u} \tag{15}$$

We then calculate clade growth rates, defined as the fold-change in relative clade frequency between time $t$ and time $t + dt$

$$\frac{\hat{x}_i(t + dt)}{x_i(t)}. \tag{16}$$

## Null models

To quantify the impact of antigenic fitness on DENV clade success, we compare our antigenically-informed model to two null models.

Under the 'equal fitness null' model, all viruses have equal total fitness (antigenic and intrinsic fitness) at all timepoints

$$f_i^{equal}(t) = 0 \tag{17}$$

$$\hat{x}_i^{equal}(t + dt) = x_i(t)e^0 = x_i(t). \tag{18}$$

Under the 'intrinsic fitness null' model, all viruses have equal antigenic fitness but serotype-specific intrinsic fitness at all timepoints

$$f_i^{intrinsic}(t) = f_0 \tag{19}$$

**Table 5.** Parameter recovery against simulated data.

| Parameter | Input value | Optimized value |
|---|---|---|
| $\beta$ | 3.25 | 3.10 |
| $\gamma$ | 0.55 | 0.56 |
| $\sigma$ | 2.35 | 2.57 |
| $f_0^{(1)}$ | 0.70 | 0.72 |
| $f_0^{(2)}$ | 0.85 | 0.78 |
| $f_0^{(3)}$ | 0.40 | 0.41 |

$$\hat{x}_i^{intrinsic}(t+dt) = x_i(t)e^{f_0}. \tag{20}$$

## Model performance assessment and parameter fitting

We assess predictive power as the root mean squared error between predicted and empirical clade frequencies. To assess both the final frequency predictions and the predicted clade trajectories, this RMSE includes error for each clade, for each starting timepoint $t$, and for each intermediate predicted timepoint $t + u$.

Our frequency prediction model has a total of six free parameters. We jointly fit these parameters to minimize RMSE of serotype frequency predictions via the Nelder-Mead algorithm as implemented in SciPy v.1.0.0 (*Table 2*) (*Jones et al., 2001*; *Gao and Han, 2012*). We use $N = 2$ years of previous immunity that contribute to antigenic fitness and project $dt = 2$ years in the future when predicting clade frequencies.

## Simulations

To ensure the model machinery functions correctly, we seeded a forward simulation of clade dynamics with 2 years of empirical frequencies and simulated predicted dynamics over the remainder of the time course (*Figure 5—figure supplement 3*). We then fit model parameters as described above, and obtained parameter values that well recover input values (*Table 5*).

## Data and software availability

Sequence and titer data, as well as all source code used for analyses and figure generation, is publicly available at https://github.com/blab/dengue-antigenic-dynamics (copy archived at https://github.com/elifesciences-publications/dengue-antigenic-dynamics). Our work relies upon many open source Python packages and software tools, including iPython (*Perez and Granger, 2007*), Matplotlib (*Hunter, 2007*), Seaborn (*Waskom, 2017*), Pandas (*McKinney, 2010*), CVXOPT (*Andersen et al., 2013*), NumPy (*van der Walt et al., 2011*; *Gao and Han, 2012*), Biopython (*Cock et al., 2009*), SciPy (*Jones et al., 2001*), Statsmodels (*Seabold and Perktold, 2010*), Nextstrain (*Hadfield et al., 2018*), MAFFT (*Katoh and Standley, 2013*), and IQ-TREE (*Nguyen et al., 2015*). Package versions are documented in the GitHub repository.

## Acknowledgements

We thank Richard Neher, John Huddleston, Andrew Rambaut, Molly OhAinle, David Shaw, Paul Edlefsen, Michal Juraska, and all members of the Bedford Lab for useful discussion and advice. SB is a Graduate Research Fellow and is supported by NSF DGE-1256082. TB is a Pew Biomedical Scholar and is supported by NIH R35 GM119774-01. LK is supported by NIH awards R01AI114703-01 and P01AI106695. Our work depends on open data sharing and many open source software tools. We gratefully acknowledge the authors and developers who make our work possible.

## Additional information

### Funding

| Funder | Grant reference number | Author |
| --- | --- | --- |
| National Science Foundation | DGE-1256082 | Sidney M Bell |
| Pew Charitable Trusts | | Trevor Bedford |
| National Institute of General Medical Sciences | R35GM119774-01 | Trevor Bedford |
| National Institute of Allergy and Infectious Diseases | R01AI114703-01 | Leah Katzelnick |
| National Institute of Allergy and Infectious Diseases | P01AI106695 | Leah Katzelnick |

The funders had no role in study design, data collection and interpretation, or the decision to submit the work for publication.

### Author contributions

Sidney M Bell, Data curation, Software, Formal analysis, Funding acquisition, Validation, Investigation, Visualization, Methodology, Writing—original draft, Writing—review and editing; Leah Katzelnick, Data curation, Methodology, Writing—review and editing; Trevor Bedford, Conceptualization, Resources, Software, Formal analysis, Supervision, Funding acquisition, Investigation, Methodology, Writing—review and editing

### Author ORCIDs

Sidney M Bell ⓘ https://orcid.org/0000-0003-1933-6033
Trevor Bedford ⓘ https://orcid.org/0000-0002-4039-5794

### Decision letter and Author response

Decision letter https://doi.org/10.7554/eLife.42496.sa1
Author response https://doi.org/10.7554/eLife.42496.sa2

## Additional files

### Supplementary files

• Transparent reporting form

### Data availability

All data, code, model implementations, analyses and figures are available via our online repository at github.com/blab/dengue-antigenic-dynamics (copy archived at https://github.com/elifesciences-publications/dengue-antigenic-dynamics).

The following datasets were generated:

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
