## [Decision Letter]

Thank you for submitting your article "Dengue antigenic relationships predict evolutionary dynamics" for consideration by *eLife*. Your article has been reviewed by Diethard Tautz as the Senior Editor, a Reviewing Editor, and three reviewers. The following individual involved in the review of your submission has agreed to reveal his identity: José Lourenço (Reviewer #3).

The reviewers have discussed the reviews with one another and the Reviewing Editor has drafted this decision to help you prepare a revised submission.

Summary:

In this manuscript, Bell and colleagues analyse antibody titer measurements from a non-human primate study and a monovalent vaccine study in humans to determine (a) whether and to what extent antigenic variation is present within dengue's 4 serotypes, DENV 1-4; and (b) whether the within-serotype antigenic variation that they find contributes to/is informative of dengue virus population dynamics.

Essential revisions:

First half of manuscript:

1) Demonstrate more effectively that the within-serotype antigenic differences are robust by:a) performing the analysis on each of the two datasets separately, indicating whether and to what extent the antigenic differences found by each of the two datasets are consistent with one another;b) providing a list of amino acid substitutions that occur on every branch with a positive inferred db value using the results derived from the use of the two datasets simultaneously.

Second half of manuscript:

2) Demonstrate the appropriateness of using strain frequencies as a proxy for serotype prevalence through a comparison with available epidemiological data.

3) Provide a demonstration on simulated data that this fitness model approach would be/would not be able to recover the "true" input model. The simulated data presumably should be at the same resolution and depth as the empirical data.

4) Given that the intrinsic fitness differences inferred are so large between the serotypes, perform an analysis that demonstrates that this null model does not do as well as the serotype-level antigenic differences (+ intrinsic fitness differences) model. Further, are these intrinsic fitness differences supported by existing epidemiological and/or virological data?

5) Incorporate uncertainty in provided estimates.

*Reviewer #1:*

In this manuscript, Bell and colleagues analyze titer measurements from a non-human primate study and a monovalent vaccine study in humans to determine (a) whether and to what extent antigenic variation is present within dengue's 4 serotypes, DENV 1-4; and (b) whether the within-serotype antigenic variation that they find contributes to/is informative of dengue virus population dynamics. The presented work uses a number of different approaches that have been previously developed elsewhere and applied mostly to influenza viruses. By applying these approaches to dengue, Bell and colleagues find evidence for within-serotype antigenic differences. These intra-serotypic antigenic differences, however, do not appear to provide better predictive power in terms of dengue's population dynamics.

Specific comments:

1) I think the authors do a good job at convincing the reader that there is antigenic variation present within dengue serotypes. However, is there already evidence for this finding based on Katzelnick's previous work using antigenic maps? If so, why is this finding novel here? I understand that a different (phylogeny-based) approach is used here, and that, for several reasons, this phylogenetic approach might be preferable over an antigenic map approach. Are the key biological results the same across these studies, though? More text on this is necessary to convince the reader that this work constitutes a major, novel result.

2) The authors infer antigenic distances based on two very different datasets: a non-human primate dataset and a human dataset. The NHP dataset uses experimental challenges with DENV, with sera being drawn 3 months post-infection. The human dataset uses sera from a monovalent vaccine study, with sera drawn after 6 weeks. Due to the difference in host species, time at which sera are drawn, and the identity of the virus (vaccine strain vs. unattenuated strain), it would be interesting to infer the antigenic relationships for each dataset separately and compare across the datasets using the phylogeny-based approach. Consistent findings across the two datasets would substantiate the results. I recognize that Katzelnick et al., 2015 compared these datasets, but they did not use the phylogeny-based approach.

3) The authors conclude that between-serotype antigenic differences are important for accurately predicting dengue dynamics, but that incorporating within-serotype antigenic variation does not provide greater prediction accuracy. Since the differences in the intrinsic fitness values of the 4 serotypes are very large (Table S1 and Figure 6), to demonstrate convincingly that antigenic differences at the between-serotype level matter, I think what must be shown is the prediction accuracy under the assumption of no antigenic variation at all. From Figure 6, it seems that intrinsic fitness differences between the 4 serotypes might account for the overwhelming majority of prediction accuracy. The authors provide a null model with no intrinsic fitness differences in the supplemental section (equations 16-19), but never report on the findings of this null model, as far as I can see. Including it in Table S1, and in text in the main manuscript, I think is important.

4) Is there a relationship between intrinsic fitnesses that are inferred in the second half of the paper and 'row and column' effects inferred in the first half of the paper? It seemed like this might be a possibility.

5) Subsection “Antigenic evolution occurs within serotypes: inference of db/lasso regression. Instead of showing Figure 5 and Figure 5—figure supplement 1, could you just show the phylogeny again (e.g. Figure 1A), with all non-zero db values that were inferred labeled on the tree? This would be a nicer, more clear visualization, I think. Along with this, could you add a table that contains, for every branch that carries a non-zero db value, the list of amino acid changes that were inferred on that branch? This should be easily do-able using Nextstrain, and it would be informative. If many of the branches with inferred non-zero db values havs no amino acid changes in env, this would also indicate possible issues with the results.

*Reviewer #2:*

The manuscript by Bell et al., performs analyses that seek to describe the structure of antigenic variation in dengue virus and to demonstrate its significance for epidemiological dynamics. The analysis of epidemiological dynamics is performed on data from Southeast Asia and considers antigenic groupings at the level of conventional serotype designations for dengue virus and also at one level below that, which are the same as conventional dengue virus genotype groupings and come out to three antigenically distinct groups below the level of each serotype. The latter set of groupings is supported by analyses presented here. One pair of these groupings below the level of serotype are consistent with a limited set of findings from dengue vaccine trials about differential efficacy by DENV-4 genotype. In terms of the influence of the antigenic groupings identified here on epidemiological dynamics, the authors found support for conventional serotype groupings being associated with patterns of serotype turnover across years. Below the level of conventional serotypes, results about the influence of antigenic groupings on epidemiological dynamics were more equivocal.

The approach used to define antigenic groupings has a direct correspondence to genetic groupings obtained through phylogenetic analysis. The estimation of how antigenic profile evolved along this phylogeny was then performed secondarily, with rates of antigenic evolution on different branches allowed to vary such that antigenic distances between available sample pairs could then be explained by summing rates of antigenic evolution across the lineages separating a given pair. This approach yielded the strongest results of the manuscript, which show that the inferred patterns of antigenic evolution along the dengue virus phylogeny explain a considerable amount of variation in antigenic distances among sample pairs (Pearson correlation = 0.86). Interestingly, this result compares favorably to a related analysis that limited antigenic evolution up to the point of the most recent common ancestor of each serotype (Pearson correlation = 0.79). As the authors acknowledge, most antigenic variation is still explained by conventional serotype groupings, but that is to be expected. And there is still a considerable amount of variation explained below the level of serotype. The use of 10-fold cross-validation is a notable strength of this analysis.

The other major component of the results pertained to the influence of antigenic evolution on epidemiological dynamics.

First, it should be noted that the basis for this analysis was a reconstruction of serotype and genotype dynamics estimated through an approach that made use of the number of lineages of a given type at each time point in the time series. Although clever, a major limitation of such an approach is that the estimated dynamics of the different types could be extremely sensitive to which samples were used to estimate the phylogeny; e.g., more samples of a given type at different times could lead to different results, and there is no guarantee that the samples were collected in some sort of representative fashion. Sampling issues such as these are major issues for many phylogenetic analyses, and I see no reason why this analysis would be exempt from that type of concern. Although there are precious few data sets that can be used to inform historical patterns of dengue serotypes, there are some based on clinical incidence that could at least be used as an independent comparison against the historical dynamics reconstructed here. Were there to be some level of agreement between these estimates of historical patterns of dengue serotypes, that could help assuage some of these concerns about sampling issues to at least some extent. One possibility is a data set from Thailand described in Reich et al.,(2013).

Second, there are two sets of results that derive from this analysis. The first is at the level of serotype, and it shows that antigenic novelty is associated with short-term changes in relative frequencies of serotypes. The second is at the level of the twelve antigenic groupings identified in the first portion of this manuscript, and it shows that resolution in antigenic variation below the level of conventional serotypes does not provide any additional power to explain short-term changes in relative frequencies of these sub-serotype groupings. In terms of the significance and implications of these two results, the first seems consistent with existing knowledge of dengue virus serotype dynamics, and the second leaves the hypothesis that sub-serotype antigenic variation affects epidemiological dynamics either unresolved (if you assume there was too much noise for this approach to work) or perhaps even refuted (if you assume that this was a fair test of that hypothesis). In light of the issues raised in my first comment about this analysis (among others), my interpretation is that there was simply not enough of a signal in the sub-serotype data to come to any firm conclusion one way or another. Truthfully, I would prefer for an analysis such as this to not even be attempted without greater confidence going into it that a valid result could be obtained (which could be demonstrated through simulation studies, for example).

*Reviewer #3:*

Summary:

The research in this manuscript describes and tries to understand immunity to / of genetic variants (genotypes) of dengue serotypes. Critically, I believe that the main questions addressed in this study could be the answer to puzzles yet to be solved on the population biology and epidemiology of dengue viruses. The results presented could not fully measure / discern the population-level impact of the newly identified 12 antigenic phenotypes versus the canonical 4, but it should motivate the community to pursue this important line of research. I found the article well written, with clear presentation of results and method explanation. I am generally happy with the results and inclined to agree with what has been demonstrated. Although I feel the results are of general interest to *eLife* readers, a wider discussion should take place on whether the manuscript is, methodologically and / or relating to data, offering enough to fit the remit of the journal. I have some specific comments, mainly related to waning, intrinsic fitness, and data / method details that I ask to be addressed by the authors.

Specific comments:

The first sentence of the Conclusions is an overstatement (an exception compared to the rest of the text, generally fair on the results presented). 'We find that within-sero antigenic evolution is necessary to explain observed patterns of cross-immunity and susceptibility…'- given the results presented, it is not true that it is 'necessary'. Perhaps a fairer statement is that '… evolution helps explain observed patterns of cross-neutralization among genotypes'. Note that cross-immunity and susceptibility are never attempted to be explained.

From the methods it seems that empirical clade frequencies were estimated from the phylogeny in Figure 1 (although not explicit), which is from a subsample (N=2563) of the entire database (N=12649). Should the frequencies be estimated from the largest set? This is important in the context of conclusion / discussion that (future) 'richer datasets' may help fill in the gaps of this study. Please clarify if only the smallest set was used and why using the largest would not have changed results of Figure 6A and Figure 7A.

The authors refer to using sequences with 'full coding sequence of E'. It is not clear to me if this means that phylogenies are based solely on ENV? (not full genome?). If so, authors should defend this decision and potentially discuss implications.

It’s important that credit is given to studies that have previously suggested that 'the null hypothesis of antigenically uniform serotypes is inconsistent with observed patterns of cross-protection and susceptibility…' (in Discussion section). The authors refer to previous study Katzelnick et al., 2015 only, but others exist, e.g. Forshey et al., 2016; Waggoner et al., 2016.

At the end of the first paragraph of the Discussion section, authors make reference to DENV4-specific and genotype dependent CYD-TDV efficacies. It isn't clear why only this example is given. E.g. it is important for the reader to know if CYD-TDV efficacies presented no differences between other genotypes, or if these were simply not measured in the trials.

Waning immunity: In the context of Figure 6, with genetic resolution and antigenic resolution at serotype and interserotype (respectively, Table S1), waning immunity (γ) can be interpreted as serotype-associated temporary transcending immunity. This is, I assume, why the estimation of fast waning makes biological sense. But in that case, I ask the authors to clarify what the interpretation of γ should be when the genetic resolution is genotype (and antigenic resolution is either interserotype or full tree; note that Table S1 has γ fixed across all models). From the definition of expression 7, does it imply that between genotype waning is as fast as between serotype waning? If so, waning exists at all (data) resolutions and is generally both transcending and temporary? This feels like a major result, but it is not discussed.

Intrinsic fitness differences (estimated values): estimations in Table S1 present significant differences between serotypes. This is not discussed in the main text and it feels like a result that should be supported by previous work. For example, are there in vitro, in vivo or population-based measures suggesting that intrinsic fitness follows the general rule DENV1 >= DENV2 >> DENV3 >> DENV4?

Intrinsic fitness differences (necessity of): the authors include the possibility of (model) intrinsic fitness differences between the serotypes. This is a reasonable assumption given the clear differences observed in empirical clade frequencies (Figure 6A), which generally follow the rule of 'success' in order DENV1 >= DENV2 >> DENV3 >> DENV4 (in fact, I believe this is an ubiquitous world trend). I would like to suggest to the authors an alternative / complementary factor that arises from the results presented (Figure 5, Figure 5—figure supplement 1). It seems that antigenic phenotypes of DENV4 are the most closely related, followed by DENV3, and DENV1-2 (visually, from branch distances between them). Could this be hinting on the fact that the higher success (empirical clade frequencies) of DENV1-2 is related to the fact that clades can more easily co-circulate given that herd-immunity escape is 'easier'? (in other words, reinfection with DENV1-2 genotypes is more common). In contrast, DENV4 would be less successful, because any clade will find high resistance to transmission (herd-immunity from other clades / antigenic phenotypes). I think it is important for the authors to consider this possibility, since the current assumption / estimation of very high differences in intrinsic fitness between serotypes may not be supported by the literature (see my other comment on Intrinsic fitness differences).

Results section: 'However, we find that accounting for within-serotype antigenic evolution substantially improves our ability to explain dengue antigenic phenotypes'. Please clarify if a more correct statement is instead: 'However, we find that accounting for within-serotype antigenic evolution substantially improves our ability to explain cross-genotype neutralization data'.

NHP versus human data: Figure 2 presents the data for 3-month post infection of NHPs. Just before the figure, this NHP data set is explained (linked to Figure 1A), but human data is also mentioned. As far as I understand the human data set is not included in this study? This is only clear when checking Figure 1A.

NHP dataset: the original data set in Katzelnick et al., 2015 (Table S4) appears to contain a larger number of sera and virus entries than presented in Figure 2 (I believe S4 is the correct dataset?). These are the discrepancies I found for virus (rows): DENV2 13 versus 16 (Figure 2, Table S4, respectively), DENV3 8 verus 9, DENV 4 9 versus 11. For sera (columns): DENV1 6 versus 8 (Figure 2, Table S4, respectively), DENV2 9 versus 12, DENV4 7 versus 8. Please clarify if I got this wrong, or if some of the data was not used in the current study.

In Figure 5, the antigenic phenotypes of sequences in Figure 1 are presented. It seems that some of the genotypes in Figure 1 are missing in the legend of Figure 5. For instance, DENV2 ASIANI, ASIANII, DENV1 II, DENV1 IV, DENV4III, etc. I understand that antigenically uniform clades have been collapsed, but the legend should include all original genotypes. Also, DENV4 sylvatic is stated, but its use and / or results associated with it have not been mentioned elsewhere in the text?

[Editors' note: further revisions were requested prior to acceptance, as described below.]

Thank you for resubmitting your work entitled "Dengue antigenic relationships predict evolutionary dynamics" for further consideration at *eLife*. Your revised article has been favorably evaluated by Diethard Tautz (Senior Editor), a Reviewing Editor, and two reviewers.

The manuscript has been improved but there are some remaining issues that need to be addressed before acceptance, as outlined below:

Title

The title must be revised. It focuses on the portion of your analysis having to do with viral clade dynamics, and the content of the statement made in this title is only supported at the serotype level. By focusing on this portion of their results – which in our opinion is secondary in its significance compared to the rest about the genetics underlying antigenic differences etc. – and not including words such as "at the serotype level". This title has great potential to be very misleading and widely misunderstood.

Abstract

"We find that antigenic fitness mediates fluctuations in DENV clade frequencies, although this appears to be primarily explained by coarser serotype-level antigenic differences." should be revised to "We find that antigenic fitness mediates fluctuations in DENV clade frequencies, but only at the serotype level."

"These results provide a more nuanced understanding of dengue antigenic evolution…" Is that really true? We suggest this statement be made more precise.

Author summary

"We find that antigenic fitness is a key determinant of DENV population turnover, although this appears to be driven by coarser serotype-level antigenic differences." should be revised to "We find that antigenic fitness is a key determinant of DENV population turnover, but only at the serotype level."

Discussion section

We wonder if there might be a third hypothesis C. Perhaps the variable cross-reactivity that they see below the serotype level impacts disease severity (as suggested in the case they built up in the introduction) but does not impair infection and subsequent transmission (which are what really have to do with viral fitness that should have been picked up on in the clade dynamics analysis). Recent work by ten Bosch et al. (PloS Pathogens, 2018) estimated that asymptomatic and mild infections are still relatively transmissible compared to more severe infections, which is consistent with this possibility.

Conclusion

"We also find that population immunity is a strong determinant of the composition of the DENV population across Southeast Asia, although this is putatively driven by coarser, serotype-level antigenic differences." should be revised to "We also find that population immunity is a strong determinant of the composition of the DENV population across Southeast Asia, but only at the serotype level."

---

## [Author Response]

Summary:In this manuscript, Bell and colleagues analyse antibody titer measurements from a non-human primate study and a monovalent vaccine study in humans to determine (a) whether and to what extent antigenic variation is present within dengue's 4 serotypes, DENV 1-4; and (b) whether the within-serotype antigenic variation that they find contributes to/is informative of dengue virus population dynamics.Essential revisions:First half of manuscript:1) Demonstrate more effectively that the within-serotype antigenic differences are robust by: a) performing the analysis on each of the two datasets separately, indicating whether and to what extent the antigenic differences found by each of the two datasets are consistent with one another;

This raises a key question of whether the identified antigenic differences between dengue viruses are relevant only for infection of non-human primates. While we agree it would be interesting to do a head-to-head comparison between the results from each dataset independently, the human dataset consists of only four serum strains. This does not provide sufficient resolution to map antigenic change to specific substitutions or branches based on the human data alone. To address the underlying question of whether these antigenic differences are species-specific, we instead examine the error distribution of out-of-sample titer predictions, stratified by serum species (Figure 2—figure supplement 1). If these inferred antigenic differences were driven by one host species, we would expect to see systematic differences in predictive power between serum isolated from each species. We observe comparable prediction error for both species, suggesting that these differences in viral antigenic phenotype are independent of host species and serum from both human and NHP fit equally well into a joint model.

b) providing a list of amino acid substitutions that occur on every branch with a positive inferred db value using the results derived from the use of the two datasets simultaneously.

As described in the cover letter, this suggestion was particularly fruitful; thank you! We now map antigenic change to substitutions, instead of branches in the phylogeny. We identify 49 specific substitutions and four colinear substitution clusters that are antigenically relevant. We report these substitutions and the magnitude of their contributions to antigenic change in Table S1 and visualize these mutations and their magnitude in new Figure 2. As described in the cover letter, properly addressing mutations lead us to switch to a “substitution” model rather than a “tree” model as primary model to estimate antigenic distances from titer data. Switching to the substitution model improved model fit over tree model (Table S2).

Second half of manuscript:2) Demonstrate the appropriateness of using strain frequencies as a proxy for serotype prevalence through a comparison with available epidemiological data.

We thank reviewer #2 for helpfully pointing us to the case count dataset from Reich et al.,2013, which consists of serotype-specific biweekly case counts for a children’s hospital in Bangkok from 1973–2010. We binned these counts into quarterly timepoints to estimate the proportion of reported cases per serotype, per quarter from 1975–2010.

We then assembled a dataset of Thai dengue sequences and estimated serotype frequencies over the same time period as described in the Methods (Equations 4–5).

We show a head-to-head comparison of these two estimates of serotype frequency in Figure 1—figure supplement 1.

As evident in the time series plots, we see generally good concordance between the two measures, although there are some time periods where our estimated frequencies do not capture all of the dengue diversity circulating (e.g., this metric ‘misses’ the early 1990s Bangkok outbreak of dengue 4 that is suggested by the case counts). We believe this is primarily due to the mismatch in sampling frame, where some of the hyperlocal transmission within Bangkok is not captured by the country-level comparison dataset.

Importantly, our estimates of serotype frequency estimates are smoothed, whereas the case counts oscillate from one time point to the next. We believe this artificially deflates correlation coefficients between the two measures.

3) Provide a demonstration on simulated data that this fitness model approach would be/would not be able to recover the "true" input model. The simulated data presumably should be at the same resolution and depth as the empirical data.

This was a very helpful suggestion. We seeded a simulation with two years of empirical serotype frequencies and then simulated forward to generate frequency values over the remainder of the time course. We then fit parameters to this simulated data as described in the Materials and methods section and were able to recover the true input parameters (Table 2, Figure 5—figure supplement 1).

4) Given that the intrinsic fitness differences inferred are so large between the serotypes, perform an analysis that demonstrates that this null model does not do as well as the serotype-level antigenic differences (+ intrinsic fitness differences) model. Further, are these intrinsic fitness differences supported by existing epidemiological and/or virological data?

In the antigenically informed model, clade fitness is additively determined by antigenic fitness (i.e., frequency-weighted antigenic distance from recently circulating clades) and intrinsic fitness (a time-invariant, serotype-specific, fit parameter).

For both serotypes and genotypes, we now also report model performance under two null formulations. In the ‘equal fitness null’ model, all clades are assigned equal total fitness. In the ‘intrinsic fitness null’, clade fitness is determined only by intrinsic fitness. As shown in Table S3, these null models all have higher prediction error than the antigenically-informed model for both serotypes and genotypes. These null models are particularly poor at predicting clade growth rates, with R^2^  ≤ 0.04. From this, it is clear that intrinsic fitness alone is a poor predictor of serotype and genotype population dynamics.

We investigated the literature and could not find data addressing these sort of intrinsic fitness differences via competition assays. We’ve made a note of this in the text:

“We also find that these dynamics are best explained by intrinsic fitness that moderately varies by serotype; we are not aware of any literature that directly addresses this observation via competition experiments.”

5) Incorporate uncertainty in provided estimates.

To assess model error, we can ask two related questions about how well it predicts the dynamics of each clade, at each starting time point:

- How well does the model predict the frequency trajectory (i.e., the change in frequency from time *t* → *t* + 0.25 → *t* + 0.5 … → *t* + *dt*) of each clade? To quantify this, we calculate the root mean squared error of frequency predictions, incorporating all stepwise predictions (of all clades, for all starting timepoints).

- How well does the model predict overall clade growth or decline (i.e., the fold-change in frequency from *t* → *t + dt*)?

To quantify this, we report accuracy (i.e., the proportion of predictions for which the predicted and observed growth rate were both > 1 or both < 1) and the correlation coefficient of predicted vs observed growth rates.

These values are now reported for all model variations in Table S3.

Reviewer #1:In this manuscript, Bell and colleagues analyze titer measurements from a non-human primate study and a monovalent vaccine study in humans to determine (a) whether and to what extent antigenic variation is present within dengue's 4 serotypes, DENV 1-4; and (b) whether the within-serotype antigenic variation that they find contributes to/is informative of dengue virus population dynamics. The presented work uses a number of different approaches that have been previously developed elsewhere and applied mostly to influenza viruses. By applying these approaches to dengue, Bell and colleagues find evidence for within-serotype antigenic differences. These intra-serotypic antigenic differences, however, do not appear to provide better predictive power in terms of dengue's population dynamics.Specific comments:1) I think the authors do a good job at convincing the reader that there is antigenic variation present within dengue serotypes. However, is there already evidence for this finding based on Katzelnick's previous work using antigenic maps? If so, why is this finding novel here? I understand that a different (phylogeny-based) approach is used here, and that, for several reasons, this phylogenetic approach might be preferable over an antigenic map approach. Are the key biological results the same across these studies, though? More text on this is necessary to convince the reader that this work constitutes a major, novel result.

The reviewer is correct that the original publication of this dataset also reported some degree of within-serotype antigenic heterogeneity. However, the models presented here add additional evidence that this observed antigenic heterogeneity is the result of an underlying evolutionary process, rather than technical noise, and that these antigenic dynamics have a significant impact on dengue evolution and population turnover. We also report a parsimonious list of substitutions that measurably impact dengue antigenic phenotypes.

2) The authors infer antigenic distances based on two very different datasets: a non-human primate dataset and a human dataset. The NHP dataset uses experimental challenges with DENV, with sera being drawn 3 months post-infection. The human dataset uses sera from a monovalent vaccine study, with sera drawn after 6 weeks. Due to the difference in host species, time at which sera are drawn, and the identity of the virus (vaccine strain vs. unattenuated strain), it would be interesting to infer the antigenic relationships for each dataset separately and compare across the datasets using the phylogeny-based approach. Consistent findings across the two datasets would substantiate the results. I recognize that Katzelnick et al., 2015 compared these datasets, but they did not use the phylogeny-based approach.

We agree that this is an important issue. Please see essential revisions response 1a.

3) The authors conclude that between-serotype antigenic differences are important for accurately predicting dengue dynamics, but that incorporating within-serotype antigenic variation does not provide greater prediction accuracy. Since the differences in the intrinsic fitness values of the 4 serotypes are very large (Table S1 and Figure 6), to demonstrate convincingly that antigenic differences at the between-serotype level matter, I think what must be shown is the prediction accuracy under the assumption of no antigenic variation at all. From Figure 6, it seems that intrinsic fitness differences between the 4 serotypes might account for the overwhelming majority of prediction accuracy. The authors provide a null model with no intrinsic fitness differences in the supplemental section (equations 16-19), but never report on the findings of this null model, as far as I can see. Including it in Table S1, and in text in the main manuscript, I think is important.

We agree that this is an important issue. Please see essential revisions response 4.

4) Is there a relationship between intrinsic fitnesses that are inferred in the second half of the paper and 'row and column' effects inferred in the first half of the paper? It seemed like this might be a possibility.

No, these are separate parameters. Virus avidity and serum potency are parameters modulating antigenic similarity between strains, separate from intrinsic fitness which affects clade success outside of antigenicity. Additionally, the ‘row and column’ effects from the first half of the paper correspond to the overall viral avidity and serum potency of specific *strains*. The intrinsic fitnesses in the second half of the paper are fit parameters that are time-invariant and correspond to *serotypes*. We apologize for the lack of clarity, and have updated the text accordingly.

5) Subsection “Antigenic evolution occurs within serotypes: inference of db/lasso regression. Instead of showing Figure 5 and Figure 5—figure supplement 1, could you just show the phylogeny again (e.g. Figure 1A), with all non-zero db values that were inferred labeled on the tree? This would be a nicer, more clear visualization, I think. Along with this, could you add a table that contains, for every branch that carries a non-zero db value, the list of amino acid changes that were inferred on that branch? This should be easily do-able using Nextstrain, and it would be informative. If many of the branches with inferred non-zero db values havs no amino acid changes in env, this would also indicate possible issues with the results.

We agree that these visualizations are more descriptive, and have added the requested table (Table S1). Please see the essential revisions response 1, as well as new Figure 2, Figure 3 and Figure 4.

Reviewer #2:The manuscript by Bell et al., performs analyses that seek to describe the structure of antigenic variation in dengue virus and to demonstrate its significance for epidemiological dynamics. The analysis of epidemiological dynamics is performed on data from Southeast Asia and considers antigenic groupings at the level of conventional serotype designations for dengue virus and also at one level below that, which are the same as conventional dengue virus genotype groupings and come out to three antigenically distinct groups below the level of each serotype. The latter set of groupings is supported by analyses presented here. One pair of these groupings below the level of serotype are consistent with a limited set of findings from dengue vaccine trials about differential efficacy by DENV-4 genotype. In terms of the influence of the antigenic groupings identified here on epidemiological dynamics, the authors found support for conventional serotype groupings being associated with patterns of serotype turnover across years. Below the level of conventional serotypes, results about the influence of antigenic groupings on epidemiological dynamics were more equivocal.The approach used to define antigenic groupings has a direct correspondence to genetic groupings obtained through phylogenetic analysis. The estimation of how antigenic profile evolved along this phylogeny was then performed secondarily, with rates of antigenic evolution on different branches allowed to vary such that antigenic distances between available sample pairs could then be explained by summing rates of antigenic evolution across the lineages separating a given pair. This approach yielded the strongest results of the manuscript, which show that the inferred patterns of antigenic evolution along the dengue virus phylogeny explain a considerable amount of variation in antigenic distances among sample pairs (Pearson correlation = 0.86). Interestingly, this result compares favorably to a related analysis that limited antigenic evolution up to the point of the most recent common ancestor of each serotype (Pearson correlation = 0.79). As the authors acknowledge, most antigenic variation is still explained by conventional serotype groupings, but that is to be expected. And there is still a considerable amount of variation explained below the level of serotype. The use of 10-fold cross-validation is a notable strength of this analysis.The other major component of the results pertained to the influence of antigenic evolution on epidemiological dynamics.First, it should be noted that the basis for this analysis was a reconstruction of serotype and genotype dynamics estimated through an approach that made use of the number of lineages of a given type at each time point in the time series. Although clever, a major limitation of such an approach is that the estimated dynamics of the different types could be extremely sensitive to which samples were used to estimate the phylogeny; e.g., more samples of a given type at different times could lead to different results, and there is no guarantee that the samples were collected in some sort of representative fashion. Sampling issues such as these are major issues for many phylogenetic analyses, and I see no reason why this analysis would be exempt from that type of concern. Although there are precious few data sets that can be used to inform historical patterns of dengue serotypes, there are some based on clinical incidence that could at least be used as an independent comparison against the historical dynamics reconstructed here. Were there to be some level of agreement between these estimates of historical patterns of dengue serotypes, that could help assuage some of these concerns about sampling issues to at least some extent. One possibility is a data set from Thailand described in Reich et al., (2013).

Thank you very much for pointing us to this dataset! We have done a head-to-head comparison as you suggested; please see essential revisions response 2.

Second, there are two sets of results that derive from this analysis. The first is at the level of serotype, and it shows that antigenic novelty is associated with short-term changes in relative frequencies of serotypes. The second is at the level of the twelve antigenic groupings identified in the first portion of this manuscript, and it shows that resolution in antigenic variation below the level of conventional serotypes does not provide any additional power to explain short-term changes in relative frequencies of these sub-serotype groupings. In terms of the significance and implications of these two results, the first seems consistent with existing knowledge of dengue virus serotype dynamics, and the second leaves the hypothesis that sub-serotype antigenic variation affects epidemiological dynamics either unresolved (if you assume there was too much noise for this approach to work) or perhaps even refuted (if you assume that this was a fair test of that hypothesis). In light of the issues raised in my first comment about this analysis (among others), my interpretation is that there was simply not enough of a signal in the sub-serotype data to come to any firm conclusion one way or another. Truthfully, I would prefer for an analysis such as this to not even be attempted without greater confidence going into it that a valid result could be obtained (which could be demonstrated through simulation studies, for example).

We share the reviewer’s frustration and thank them for the comment. We simulated frequencies over time and fit model parameters as described, demonstrating that the model is able to recover the true input parameters. The inability of a null result to distinguish between lack of evidence and lack of effect is a classic scientific challenge.

We’ve revised the text to highlight this exact issue:

“Overall, we find that antigenic fitness explains a moderate portion of the observed variation in genotype growth and decline. Surprisingly, however, we find that incorporating within-serotype antigenic differences does not improve our predictions (Figure 6C-D). This suggests that although genotypes are antigenically diverse, these differences do not appear to influence large-scale regional dynamics over time. This lack of signal could be explained by either (A) genotype- level frequency trajectories estimated from public data are overly noisy for this application or (B) our model of antigenic fitness based on PRNT assay data does not match reality, due to either PRNT assay data not well reflecting human immunity or due to our particular model formulation that parameterizes immunity from titer distances (Eq. 6–9). In the present analysis, we are not able to firmly resolve these disparate possibilities.”

We’ve tried to be as *honest* as possible with our attempt to tackle within-serotype dengue heterogeneity. We believe that there are real PRNT differences between genotypes within a serotype, where we have extended Katzelnick et al., (2015) results in showing an agreement between titers and genetic relations among viruses. We did our best to construct a model of clade dynamics that incorporates antigenic relationships. We believe presenting the null result here (caveated as it is) is the most honest thing we can do in this situation. But we agree there’s ample opportunity to improve a fitness model of clade dynamics. We’ve provided all source code in effort to encourage other groups to attempt this.

Reviewer #3:Summary:The research in this manuscript describes and tries to understand immunity to / of genetic variants (genotypes) of dengue serotypes. Critically, I believe that the main questions addressed in this study could be the answer to puzzles yet to be solved on the population biology and epidemiology of dengue viruses. The results presented could not fully measure / discern the population-level impact of the newly identified 12 antigenic phenotypes versus the canonical 4, but it should motivate the community to pursue this important line of research. I found the article well written, with clear presentation of results and method explanation. I am generally happy with the results and inclined to agree with what has been demonstrated. Although I feel the results are of general interest to eLife readers, a wider discussion should take place on whether the manuscript is, methodologically and / or relating to data, offering enough to fit the remit of the journal. I have some specific comments, mainly related to waning, intrinsic fitness, and data / method details that I ask to be addressed by the authors.Specific comments:The first sentence of the Conclusions is an overstatement (an exception compared to the rest of the text, generally fair on the results presented). 'We find that within-sero antigenic evolution is necessary to explain observed patterns of cross-immunity and susceptibility…'- given the results presented, it is not true that it is 'necessary'. Perhaps a fairer statement is that '… evolution helps explain observed patterns of cross-neutralization among genotypes'. Note that cross-immunity and susceptibility are never attempted to be explained.

We agree that the original paragraph was an overstatement. We’ve made the suggested change.

From the methods it seems that empirical clade frequencies were estimated from the phylogeny in Figure 1 (although not explicit), which is from a subsample (N=2563) of the entire database (N=12649). Should the frequencies be estimated from the largest set? This is important in the context of conclusion / discussion that (future) 'richer datasets' may help fill in the gaps of this study. Please clarify if only the smallest set was used and why using the largest would not have changed results of Figure 6A and Figure 7A.

This was a good suggestion. As explained in the cover letter, we now use a dataset of all available Southeast Asian sequences in our frequency estimations and report similar results.

The authors refer to using sequences with 'full coding sequence of E'. It is not clear to me if this means that phylogenies are based solely on ENV? (not full genome?). If so, authors should defend this decision and potentially discuss implications.It’s important that credit is given to studies that have previously suggested that 'the null hypothesis of antigenically uniform serotypes is inconsistent with observed patterns of cross-protection and susceptibility…' (in Discussion). The authors refer to previous study Katzelnick et al., 2015 only, but others exist, e.g. Forshey et al., 2016; Waggoner et al., 2016.

Thank you for catching this! We have added these references as suggested.

At the end of the first paragraph of the Discussion, authors make reference to DENV4-specific and genotype dependent CYD-TDV efficacies. It isn't clear why only this example is given. E.g. it is important for the reader to know if CYD-TDV efficacies presented no differences between other genotypes, or if these were simply not measured in the trials.

We agree this was confusing. We no longer reference the genotype analyses of the CYD-TDV trials, and instead provide a nuanced picture of predicted cross-protection based on each of the four monovalent components of the NIH vaccine candidate in new Figure 3. We felt this was a more useful discussion of the potential impact of antigenic variation on vaccine efficacy, given that these strains are in our dataset and can be examined directly.

Waning immunity: In the context of Figure 6, with genetic resolution and antigenic resolution at serotype and interserotype (respectively, Table S1), waning immunity (γ) can be interpreted as serotype-associated temporary transcending immunity. This is, I assume, why the estimation of fast waning makes biological sense. But in that case, I ask the authors to clarify what the interpretation of γ should be when the genetic resolution is genotype (and antigenic resolution is either interserotype or full tree; note that Table S1 has γ fixed across all models). From the definition of expression 7, does it imply that between genotype waning is as fast as between serotype waning? If so, waning exists at all (data) resolutions and is generally both transcending and temporary? This feels like a major result, but it is not discussed.

We originally fit model parameters separately for each clade and antigenic resolution. However, upon further reflection, we think it is more appropriate to compare model performance across the same set of parameters. We now fit parameters to serotype clade dynamics and use these parameters throughout.

The reviewer is correct that our primary model assumes no fundamental difference between homotypic reinfection and heterotypic reinfection; everything is based on PRNT titer, wherein homotypic infections tend to be more antigenically similar than heterotypic infections according to PRNT. Our model of waning would indeed imply that homotypic waning is as fast as heterotypic waning but starts from a higher baseline of immunity. This is consistent with dynamics reported in (Katzelnick et al., 2018, Figure 1C).

We now compare the performance of this model to a variant wherein homotypic immunity does not wane, but heterotypic immunity wanes exponentially. We find approximately equal model performance under these two formulations of waning (Table 2).

We’ve made a note in the text highlighting this assumption of the primary model and the comparison between the two variants:

Results section

“This model assumes no fundamental difference between homotypic and heterotypic reinfection; rather, homotypic immunity is assumed to wane at the same rate as heterotypic immunity, but starts from a higher baseline of protection based on closer antigenic distances.”

Discussion section

“Our primary model parameterization assumes that both heterotypic and homotypic immunity wane linearly over time at the same rate, with homotypic immunity starting from a higher baseline of protection based on closer antigenic distances. We compared this to a secondary model parameterization with only heterotypic waning (see Materials and methods), under which we observe similar model performance (Table 2).”

Intrinsic fitness differences (estimated values): estimations in Table S1 present significant differences between serotypes. This is not discussed in the main text and it feels like a result that should be supported by previous work. For example, are there in vitro, in vivo or population-based measures suggesting that intrinsic fitness follows the general rule DENV1 >= DENV2 >> DENV3 >> DENV4?

Please see essential revisions response 4.

Intrinsic fitness differences (necessity of): the authors include the possibility of (model) intrinsic fitness differences between the serotypes. This is a reasonable assumption given the clear differences observed in empirical clade frequencies (Figure 6A), which generally follow the rule of 'success' in order DENV1 >= DENV2 >> DENV3 >> DENV4 (in fact, I believe this is an ubiquitous world trend). I would like to suggest to the authors an alternative / complementary factor that arises from the results presented (Figure 5, Figure 5—figure supplement 1). It seems that antigenic phenotypes of DENV4 are the most closely related, followed by DENV3, and DENV1-2 (visually, from branch distances between them). Could this be hinting on the fact that the higher success (empirical clade frequencies) of DENV1-2 is related to the fact that clades can more easily co-circulate given that herd-immunity escape is 'easier'? (in other words, reinfection with DENV1-2 genotypes is more common). In contrast, DENV4 would be less successful, because any clade will find high resistance to transmission (herd-immunity from other clades / antigenic phenotypes). I think it is important for the authors to consider this possibility, since the current assumption / estimation of very high differences in intrinsic fitness between serotypes may not be supported by the literature (see my other comment on Intrinsic fitness differences).

This is an interesting hypothesis! We noticed the same trend in empirical clade frequencies. We believe this hypothesis makes intuitive sense, given what we observe in new Figure 4, where DENV2 has the most within serotype diversity. Although we’d note that DENV1 and DENV3 appear similar to have similar amount of within serotype antigenic diversity as DENV4 according to PRNT titers. More importantly, the fitness model provides exactly this possibility. Differences in antigenic phenotype among DENV2 genotypes should promote coexistence. This can be seen in new Figure 5 where the spike in DENV2 Asian I frequency in 1994 does not uniformly crash DENV2 fitness across genotypes. But despite this, we still observe improved model performance when including a serotype-specific fitness effect.

Results section: 'However, we find that accounting for within-serotype antigenic evolution substantially improves our ability to explain dengue antigenic phenotypes'. Please clarify if a more correct statement is instead: 'However, we find that accounting for within-serotype antigenic evolution substantially improves our ability to explain cross-genotype neutralization data'.

Thank you for the suggestion; we have removed this sentence.

NHP versus human data: Figure 2 presents the data for 3-month post infection of NHPs. Just before the figure, this NHP data set is explained (linked to Figure 1A), but human data is also mentioned. As far as I understand the human data set is not included in this study? This is only clear when checking Figure 1A.

We use both NHP and human monovalent data in our study. We have clarified the figure legend and the Results section ‘Dengue neutralization titer data’.

NHP dataset: the original data set in Katzelnick et al., 2015 (Table S4) appears to contain a larger number of sera and virus entries than presented in Figure 2 (I believe S4 is the correct dataset?). These are the discrepancies I found for virus (rows): DENV2 13 versus 16 (Figure 2, Table S4, respectively), DENV3 8 verus 9, DENV 4 9 versus 11. For sera (columns): DENV1 6 versus 8 (Figure 2, Table S4, respectively), DENV2 9 versus 12, DENV4 7 versus 8. Please clarify if I got this wrong, or if some of the data was not used in the current study.

We have edited the Materials and methods section on titer data to clarify that we first normalize all titer measurements and then average across individuals to yield one value of antigenic distance for each pair of (virus strain, serum strain).

In Figure 5, the antigenic phenotypes of sequences in Figure 1 are presented. It seems that some of the genotypes in Figure 1 are missing in the legend of Figure 5. For instance, DENV2 ASIANI, ASIANII, DENV1 II, DENV1 IV, DENV4III, etc. I understand that antigenically uniform clades have been collapsed, but the legend should include all original genotypes. Also, DENV4 sylvatic is stated, but its use and / or results associated with it have not been mentioned elsewhere in the text?

This figure has been replaced by new Figure 3 and Figure 4, as suggested by reviewer #2, and genotypes directly labeled in Figure 4.

[Editors' note: further revisions were requested prior to acceptance, as described below.]

The manuscript has been improved but there are some remaining issues that need to be addressed before acceptance, as outlined below:TitleThe title must be revised. It focuses on the portion of your analysis having to do with viral clade dynamics, and the content of the statement made in this title is only supported at the serotype level. By focusing on this portion of their results – which in our opinion is secondary in its significance compared to the rest about the genetics underlying antigenic differences etc. – and not including words such as "at the serotype level". This title has great potential to be very misleading and widely misunderstood.

We now see the concern that the title could be misinterpreted (certainly not our intention), and greatly appreciate the reviewer’s suggestion. We have revised the title to be more specific.

Abstract"We find that antigenic fitness mediates fluctuations in DENV clade frequencies, although this appears to be primarily explained by coarser serotype-level antigenic differences." should be revised to "We find that antigenic fitness mediates fluctuations in DENV clade frequencies, but only at the serotype level."

We agree this sentence was unclear. We have revised it to emphasize that we find evidence that serotype-level antigenic fitness is a driver of clade dynamics:

“…and find that serotype-level antigenic fitness is a key determinant of DENV clade turnover.”

We appreciate the reviewer’s contribution of suggested wording and their preservation of sentence structure. However, we thought it was unclear whether “serotype-level” modified “antigenic fitness” or “clade frequencies,” and have slightly modified the suggested wording throughout. We appreciate and welcome any further suggestions.

"These results provide a more nuanced understanding of dengue antigenic evolution…" Is that really true? We suggest this statement be made more precise.

We agree that this sentence needed to be more specific, and have revised it accordingly:

“By leveraging both molecular data and real-world population dynamics, these results provide a more nuanced understanding of the relationship between dengue genetic and antigenic evolution, and quantify the effect of antigenic fitness on dengue evolutionary dynamics.”

Author summary"We find that antigenic fitness is a key determinant of DENV population turnover, although this appears to be driven by coarser serotype-level antigenic differences." should be revised to "We find that antigenic fitness is a key determinant of DENV population turnover, but only at the serotype level."

We have revised this sentence to read:

“We find that serotype-level antigenic fitness is a key determinant of DENV population turnover.”

Discussion sectionWe wonder if there might be a third hypothesis C. Perhaps the variable cross-reactivity that they see below the serotype level impacts disease severity (as suggested in the case they built up in the introduction) but does not impair infection and subsequent transmission (which are what really have to do with viral fitness that should have been picked up on in the clade dynamics analysis). Recent work by ten Bosch et al. (PLoS Pathogens, 2018) estimated that asymptomatic and mild infections are still relatively transmissible compared to more severe infections, which is consistent with this possibility.

This is an interesting hypothesis! We note that this is also consistent with the observations of Nagao and Koelle (*PNAS,* 2008) that dengue epidemiological dynamics are consistent with a model wherein immunity confers protection from symptomatic infection but does not inhibit asymptomatic infection or onward transmission. We have added a paragraph to the Discussion section:

“….Surprisingly, however, we find that incorporating within-serotype antigenic differences does not improve our predictions (Figure~\ref{genotype_fitness}C-D). We suggest two possible explanations for this observation.

First, it may be that although genotypes are antigenically diverse, these differences do not influence large-scale regional dynamics over time. We may then hypothesize that within-serotype antigenic heterogeneity mediates disease severity, but does not influence infection or onward transmission. This hypothesis is consistent with the findings of \citet{nagao2008decreases}, who demonstrated that dengue epidemiological dynamics are compatible with a model wherein immunity confers protection against severe symptoms, but not asymptomatic infection. This is also consistent with \citet{tenBosch2018contributions}'s findings that asymptomatic dengue infections contribute to onward transmission.”

Conclusion"We also find that population immunity is a strong determinant of the composition of the DENV population across Southeast Asia, although this is putatively driven by coarser, serotype-level antigenic differences." should be revised to "We also find that population immunity is a strong determinant of the composition of the DENV population across Southeast Asia, but only at the serotype level."

We have revised this sentence to:

“We also find that serotype-level population immunity is a strong determinant of viral clade dynamics across Southeast Asia.”